# Interpreting tree ensemble machine learning models with endoR

**Albane Ruaud**[1], **Niklas Pfister**[2], **Ruth E. Ley**[1], **Nicholas D. Youngblut**[1]*

**1** Department of Microbiome Science, Max Planck Institute for Developmental Biology, Tuebingen, Germany,
**2** Department of Mathematical Sciences, University of Copenhagen, Copenhagen, Denmark

* nicholas.youngblut@tuebingen.mpg.de

**Data Availability Statement:** All data sets, analysis and results scripts are available at https://github. com/aruaud/endoR_data_analysis and https:// figshare.com/projects/Ruaud2022_endoR/142265. Our method has been fully implemented as an R-

## Abstract

Tree ensemble machine learning models are increasingly used in microbiome science as they are compatible with the compositional, high-dimensional, and sparse structure of sequence-based microbiome data. While such models are often good at predicting phenotypes based on microbiome data, they only yield limited insights into how microbial taxa may be associated. We developed endoR, a method to interpret tree ensemble models. First, endoR simplifies the fitted model into a decision ensemble. Then, it extracts information on the importance of individual features and their pairwise interactions, displaying them as an interpretable network. Both the endoR network and importance scores provide insights into how features, and interactions between them, contribute to the predictive performance of the fitted model. Adjustable regularization and bootstrapping help reduce the complexity and ensure that only essential parts of the model are retained. We assessed endoR on both simulated and real metagenomic data. We found endoR to have comparable accuracy to other common approaches while easing and enhancing model interpretation. Using endoR, we also confirmed published results on gut microbiome differences between cirrhotic and healthy individuals. Finally, we utilized endoR to explore associations between human gut methanogens and microbiome components. Indeed, these hydrogen consumers are expected to interact with fermenting bacteria in a complex syntrophic network. Specifically, we analyzed a global metagenome dataset of 2203 individuals and confirmed the previously reported association between *Methanobacteriaceae* and *Christensenellales*. Additionally, we observed that *Methanobacteriaceae* are associated with a network of hydrogen-producing bacteria. Our method accurately captures how tree ensembles use features and interactions between them to predict a response. As demonstrated by our applications, the resultant visualizations and summary outputs facilitate model interpretation and enable the generation of novel hypotheses about complex systems.

## Author summary

Machine learning models have proven to be successful at predicting diseases and other human phenotypes from microbiome data; however, gaining insight from such complex models is often challenging. To this end, we developed endoR, an R-package for enhanced

package named endoR and is available for download and with its manual at GitHub (https://github.com/leylabmpi/endoR) under an MIT license.

**Funding:** This work was supported by the Max Planck Society to AR, NY, and RL. NP was supported by a research grant (0069071) from Novo Nordisk Fonden. The funders had no role in study design, data collection and analysis, decision to publish, or preparation of the manuscript.

**Competing interests:** The authors have declared that no competing interests exist.

interpretation of tree ensemble models (e.g., random forests), the most popular and highest-performing machine learning models applied to microbiome data to date. Our method simplifies models and extracts information on associations between microbiome data, host metadata and covariates, and a predicted trait (e.g., disease versus healthy). endoR has two main strengths: i) the ability to capture interactions between predictors, and ii) regularization steps that avoid overfitting. Through extensive validations, we show that endoR is comparable in accuracy to other common approaches while easing and enhancing model interpretation. We applied endoR to gain insight into a complex syntrophic network of human gut methanogens and bacterial fermenters. Overall, endoR is a powerful tool for gaining insight from tree ensemble models applied to microbiome data.

This is a *PLOS Computational Biology* Methods paper.

## Introduction

The gut microbiome plays critical roles in many aspects of human physiology, such as digestion, immunity, and development [1–3], and has been implicated in a number of diseases [4] such as inflammatory bowel disease (IBD) [5, 6], obesity [7, 8], diabetes [9], and cancer [10, 11]. The low cost of fecal microbiome sequencing allows researchers and clinicians to relate disease states to microbiome data and also to investigate possible microbial involvement in disease [12–14].

Machine Learning (ML) models have been shown to accurately predict human host phenotypes from gut microbiome taxonomic and genomic data [15–18]. While the complexity of these models can capture interactions between variables in such data, it also complicates their interpretation. This consequently limits insights into relationships between the microbiome and human characteristics. Random forest (RF) models [19], a type of tree ensemble model, often achieve the best accuracies for predictions made with microbiome data [15–18]. A RF consists of a combination of decision trees. Each partitions all observations into subsamples with similar response values, based on a set of features. For example, a decision tree may show that diseased individuals generally have high abundances of microbes A and B, but low abundances of microbe C. Hundreds of decision trees are built from random subsamples of features and observations and aggregated to make predictions. This procedure is called bootstrap aggregation or bagging [19]; it generally leads to high accuracies with less overfitting but increases model complexity [20].

The model complexity can be mitigated by reducing the number of features via feature selection: the pre-selection of relevant features to include in the final model [21–24]. These pre-selection approaches are often based on different measures of how important a specific feature is for the prediction, e.g., Gini and permutation importances [19, 25], though many others exist [26–30]. In our framework we consider the feature selection part of the model fitting (see S1(A) Fig for details).

Feature selection methods themselves can also be used directly for model interpretation, and recently developed feature importance methods are gaining popularity within microbiome science. For instance, Ai et al. [26] utilized feature selection by mutual information to identify specific microbes predictive of colorectal cancers. Alternatively, Gou et al. [31] used SHapley Additive exPlanations (SHAP, [28]) to select microbiome features associated with type 2 diabetes that they then correlated to host genetics and risk factors using generalized linear models.

Shapley values measure the contribution of variables to the prediction of each observation [32] and can be estimated through various methods [33]. For instance, the SHAP method additively decomposes predictions into separate parts corresponding to each variable [28]. As Shapley values generate local, per-observation interpretations, they generally do not address the question of the global associations of features with the response [33]. Furthermore, SHAP makes the assumption that variables have additive effects, although tree ensembles are not additive models, therefore resulting in potentially biased estimates of feature interactions [34]. Finally, SHAP interactions are calculated only for pairs of variables, rendering their interpretation challenging for high-dimensional data sets [35].

Individual decision trees can inform on variable interactions associated with predictions. Variables belonging to a same tree branch are used in concert to make predictions; they are thus more likely to be jointly associated with the response compared to variables never appearing in the same branch [36]. However, unimportant variables may occur along decision paths, given that tree ensembles, such as RFs, are generated with a greedy procedure. To remove noise and facilitate the interpretation of tree ensembles, Friedman et al. [37] propose to remove unimportant variables from decision paths via lasso regression and thus create surrogate models to tree ensembles. The inTrees R-package [38] and random intersection tree algorithm [27] implement similar ideas of simplifying tree ensembles to obtain a reduced set of decisions from a forest. However, they lack the tools to interpret the simplified decisions further. Conversely, the randomForestExplainer R-package [39] measures variable interactions by counting the number of co-occurrences of features in decision trees. However, noise is not removed from tree ensembles before measuring variable co-occurrences. The package also does not generate easy to interpret results for models fitted on high-dimensional data.

To better interpret fitted tree ensemble models, we developed endoR, a framework for interpreting tree ensemble models. endoR utilizes decisions extracted from fitted models to infer associations between features and measures their contribution to the decision ensemble (Fig 1). The endoR workflow consists of extracting all decisions from a tree ensemble model, simplifying them, and then calculating the importance and influence of variables and interactions among pairs of variables. More specifically, the importance measures the gain in predictive accuracy attributed to a variable (or an interaction among a pair of variables), while the influence measures how the inclusion of a variable (or an interaction among a pair of variables) changes model predictions. Results are displayed as multiple intelligible plots to enhance the readability of feature and interaction importances and influences.

Notably, endoR generates a decision network which visualizes the fitted model as follows: (i) nodes represent the features used in the model, with their size and color encoding the feature importance and influence (i.e., the strength and direction of association with the response, respectively); (ii) edges represent interaction effects on the response between two features. Similarly, the width and color encodes the interaction importance and influence, respectively.

We benchmarked endoR on both a fully simulated data set and a real metagenome data set [40], both with an artificially generated phenotype as response. In particular, we compared endoR with state-of-the-art procedures commonly used for analyzing microbiome data. Altogether, our results showed that endoR successfully extracts complex interactions from tree ensemble models and performs better or comparable to existing methods. We then employed endoR on a metagenome dataset published by Qin et al. [41], in which the original study identified certain gut microbiome features to be associated with cirrhosis. From a single application of endoR, we were able to recover all major results of the original study and expand upon them by identifying additional oral bacteria colonizing the gut of patients with cirrhosis and the depletion of bacteria associated with healthy microbiome [42]. Finally, we used endoR to explore patterns of gut microbial relative abundances predictive of the presence of

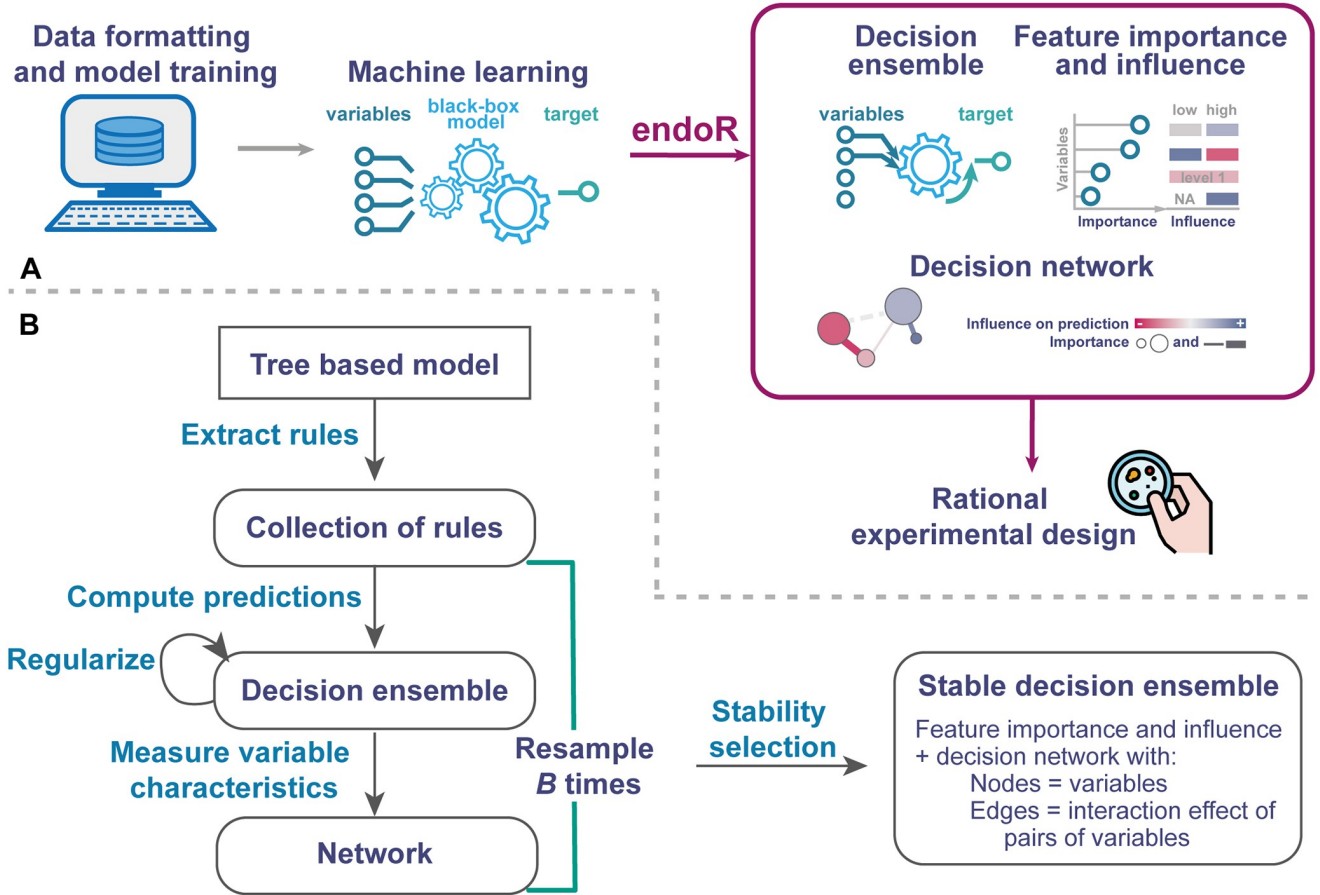

**Fig 1. Description of the endoR method workflow.** A: General overview of the workflow from data acquisition to the visualization of a network. endoR is applied to a trained classification or regression tree ensemble model. The model is first simplified into a decision ensemble, which is used to calculate the feature importance and influence on predictions. The resultant metrics are displayed in a summary plot listing the feature importance and influence, and as a decision network. The decision network illustrates the association between the response and single or pairs of variables, in regards to feature importance and influence. If the influence of a variable depends on other variables, it will be visible in the network via edges between these nodes. B: Steps taken by endoR to generate a stable network. endoR accepts tree ensemble models that were made with the XGBoost, gbm, randomForest or ranger R-packages [43–46]. Regularization is optional and consists of simplifying decisions and the decision ensemble to reduce noise. The procedure can be repeated on $B$ bootstraps to select stable decisions prior to constructing the final network.

*Methanobacteriaceae* in human guts. The presence of these methanogens was strongly associated with the presence of members of the *CAG-138* family (order *Christensenellales*), specifically with the *Phil-1* genus, as well as with members of the *Oscillospirales* order. Moreover, host traits such as the body mass index (BMI), were not predictive of the presence of *Methanobacteriaceae*, suggesting that the microbiome composition primarily determines *Methanobacteriaceae* prevalence across human populations. Taken together, the application of endoR provides new perspectives on the prevalence of *Methanobacteriaceae* in the human gut and their plausible interactions with members of the gut microbiome.

## Results

### Interpreting tree ensemble models with endoR

Tree ensemble models are often used in microbiome science, although their interpretation is limited by their complexity. endoR overcomes this issue by taking a fitted model as input and

visualizing the most relevant parts of the model in a feature importance and influence plot and a decision network. It is implemented as an R-package and accepts fitted RF and gradient boosted tree models (both for regression and classification tasks) generated using the XGBoost, gbm, randomForest, or ranger R-packages [43–46]. Note that endoR can be applied to any type of structured data (e.g., relative abundances from metagenomes or 16S data, cell counts, covariates, etc.) due to the use of tree ensemble models. Nonetheless, endoR does not explicitly take into account the compositional nature of sequencing data. We illustrate the use of endoR on a data set consisting of 2147 human gut metagenomes, with relative abundances of 520 taxa (including species, genus, and family taxonomic rank), and an artificial phenotype. The phenotype was a binary response variable taking the values '-1' or '1', it was simulated using 9 randomly selected taxa and a randomly generated categorical variable separating samples in 4 groups (labelled a, b, c, and d). Selected taxa could be from the species, genus, or family taxonomic levels to mirror the range of interactions that can occur in reality between microbial clades of varying taxonomic resolution. The mechanism generating the artificial phenotype, which we use as response variable in the classification below, is detailed in Methods and Table 1; the taxa associated with the artificial phenotype within each group are visualized in Fig 2A–2F. Our goal is to recover as much information about the mechanism generating the artificial phenotype as possible. For example, is it possible to determine that in Group a, high relative abundances of *Alistipes A*, and high relative abundances of *Marvinbryantia sp900066075* lead to a positive value of the artificial phenotype? Even though Fig 2A–2F may suggest that this is an easy classification task, it is in fact highly non-trivial; the simulation involves high order interactions (up to order 4) in a high-dimensional setting (521 variables) with strong dependencies between the features from real metagenomes. Tree ensemble models such as RFs excel in these settings, but they do not provide methods for extracting complex information about the model. This is the gap that endoR aims to fill.

First, we fit a model that predicts the artificial phenotype from the taxa (i.e., the relative abundances of species, genera, and families) and the 'group' variable. Given the high number

**Table 1. Predetermined decision rules based on the making of the artificial phenotypes.**

| Decision rule | Response |
|---|---|
| Group = 'a' & $t_1 > 0$ & $t_2 > 0$ | 1 |
| Group = 'a' & $t_1 \leq 0$ | -1 |
| Group = 'a' & $t_2 \leq 0$ | -1 |
| Group = 'b' & $t_3 > 0.1$ & $t_4 > 0.003$ | 1 |
| Group = 'b' & $t_3 \leq 0.1$ | -1 |
| Group = 'b' & $t_4 \leq 0.003$ | -1 |
| Group = 'c' & $t_5 > 0$ & $t_6 > 0$ & $t_7 > 0.01$ | 1 |
| Group = 'c' & $t_7 > 0.01$ | 1 |
| Group = 'c' & $t_5 > 0$ & $t_6 > 0$ | 1 |
| Group = 'c' & $t_5 \leq 0$ & $t_6 \leq 0$ & $t_7 \leq 0.01$ | -1 |
| Group = 'c' & $t_5 \leq 0$ & $t_7 \leq 0.01$ | -1 |
| Group = 'c' & $t_6 \leq 0$ & $t_7 \leq 0.01$ | -1 |
| Group = 'd' & $t_8 \leq 3.98 \cdot 10^{-4}$ & $t_9 > 0$ | 1 |
| Group = 'd' & $t_8 > 3.98 \cdot 10^{-4}$ | -1 |
| Group = 'd' & $t_9 \leq 0$ | -1 |

$t_i$, $i \in \{1, \ldots, 9\}$ were randomly sampled from the variables of the metagenomes detected in more than 50% of the samples.

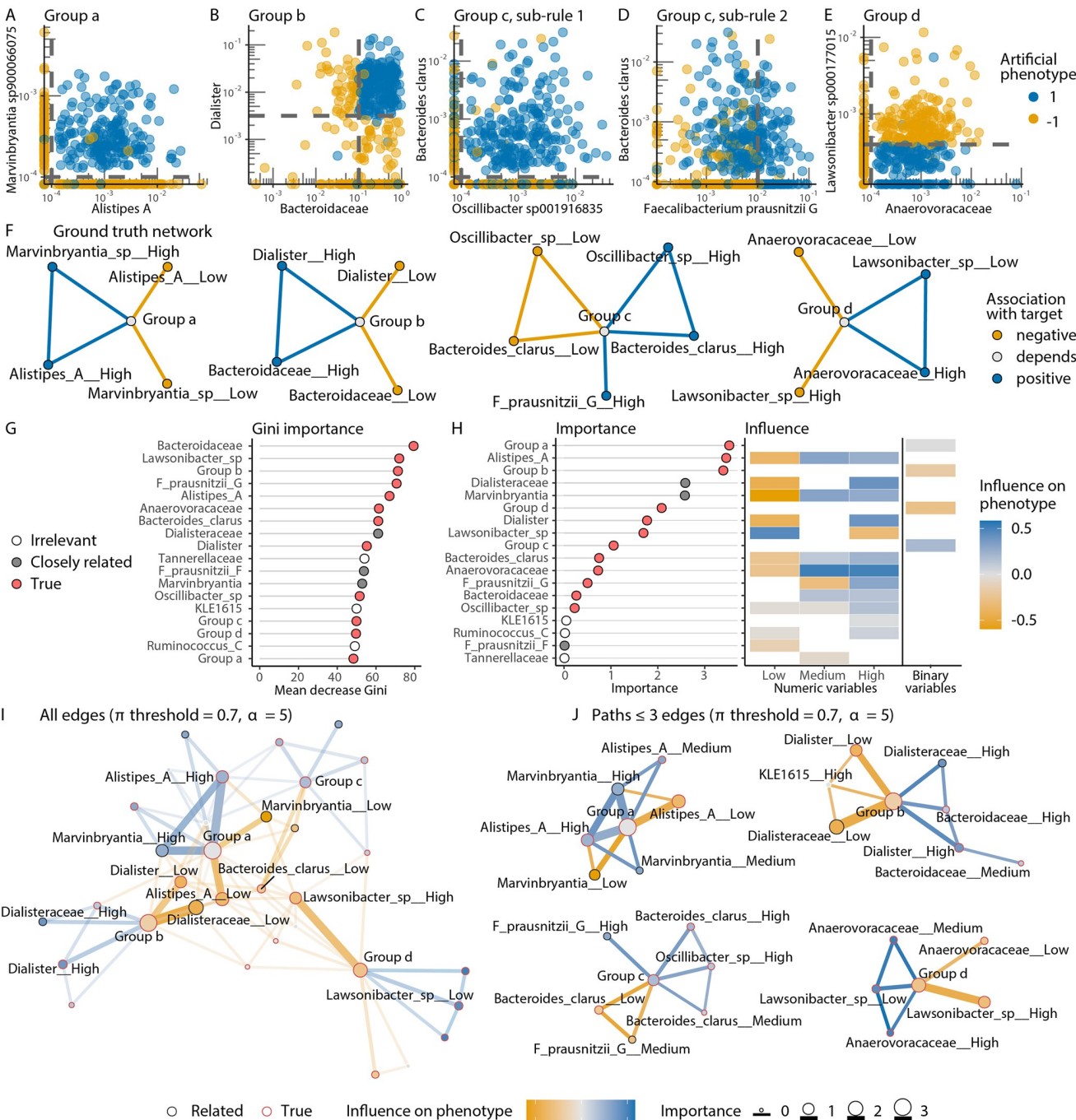

**Fig 2. endoR captures interactions predictive of an artificial phenotype from a random forest fitted on real metagenomes.** A-E: Real metagenomes with an artificial phenotype (AP): samples were separated into 4 groups (labelled a-d), a binary response variable ('1' = blue, '-1' = yellow) was simulated so that it could be predicted from a set of decisions based on the 'group' categorical feature and specific, randomly chosen microbial abundance features (e.g., 'Alistipes A'). Dashed grey lines denote thresholds in the predetermined decisions used to make the response variable and are described in Table 1 (e.g., the response variable is '1' if samples belong to Group a and have non-null relative abundances of both *Alistipes A* and *Marvinbryantia sp900066075*). For samples in Group c, the response variable was built with an 'OR' rule (i.e., 'Group = c & ((B. clarus >0 & Oscillibacter sp001916835 >0) | F. prausnitzii G >10$^{-2}$)'), so each of the two sub-rules are shown in C and D. F: Ground truth network of features derived from the response variable generation procedure described in A. Pairs of variables predicting '1' are linked by a blue edge ('positive') and those predicting '-1' by a yellow edge ('negative'). Variables for which high values are predictive of '1' have a blue node color ('positive') and a yellow node color if high values are predictive of '-1' ('negative'). If high values are predictive of '1' or '-1' depending of other variable values (e.g., Group b predicts '1' if V3 takes high values, but '-1' if V3 has low values), the color is grey ('depends'). G-H: Variable importances from the RF model as measured by the mean decrease in Gini impurity and endoR. Due to the feature

selection step, the RF model was fitted on the 18 selected features shown on the y-axis; the feature importance of all other taxa can be considered null for both. The point color indicates whether the features were used to construct the response ('True') and those taxonomically related to them ('closely related'), with 'closely related' defined as the immediate parent or child taxonomic classification in the taxonomy hierarchy (e.g., the *Bacteroides* genus is the child of the *Bacteroidaceae* family, while *Bacteroidaceae* is the parent of *Bacteroides*). I: Full decision network extracted by endoR from a RF model trained on the dataset described in A. Only the 20 features with the highest feature importance are labelled. The edge transparency is inversely proportional to the importance for I: only. J: Same network as shown in I, but edges with lowest interaction importance were removed to obtain paths between nodes of length ≤ 3. All features are labelled.

of features, we use an RF with feature selection (see Methods; Artificial phenotypes). The fitted model in this case has a cross-validation (CV) generalization error of 85.19±2.36 for the accuracy and 0.70±0.05 for Cohen's $\kappa$. Next, we apply endoR, which outputs two plots: an importance and influence plot (Fig 2H) and a decision network (Fig 2I and 2J).

The importance and influence plot shows the feature importance and influence for a single variable (Fig 2H). The importance measures how much a single variable improves the overall prediction of the model; it is similar to other well-established importance measures such as the Gini importance (given in Fig 2G) but is more accurate in its ranking of variables, such that irrelevant taxa were given the lowest feature importance by endoR but not by the Gini importance (Fig 2G and 2H). As shown in our simulations below, the endoR feature importance improves on standard importance measures for tree ensemble models. As a complement, the influence measures the change in predicted value due to the variable. For binary features, it indicates whether samples falling in that category take on average higher or lower response variable values. For instance, Fig 2H shows that samples from Group d are more likely to have a '-1' artificial phenotype (orange) while there is no clear association for samples in Group a (grey). Hence, Group a is important for predictions, but its association with the artificial phenotype may mostly depend on other features. For numeric features (taxa), the influence similarly shows how the variable is associated with the response in the final decision ensemble. To help readability, numeric variables are split into levels defined by value ranges. The number of levels is pre-specified by the user; here, 'low', 'medium', and 'high' values of each variable are assessed. If a level does not appear in the decision ensemble, its influence cannot be calculated, and so it is left blank in the plot. Fig 2H shows that 'low' relative abundances of *Alistipes A* are associated with the '-1' phenotype while 'medium' and 'high' relative abundances of this taxon are associated with the '1' phenotype. The importance and influence plot thus provides an overview of important features and how they affect the response on average.

The decision network allows for a more detailed analysis. The nodes in the network correspond to each possible value of categorical features (e.g., 'Group a') and levels of numerical features (the level is indicated by '__Level', e.g., 'Marvinbryantia__High'). The size of the node corresponds to the importance, while the color encodes the influence. Edges correspond to interaction effects, while size and color indicate importance and influence of the interaction, respectively. Either the full network can be displayed (Fig 2I) or only the most important paths composed by less than three edges (Fig 2J). For example, in Fig 2J, we can see that the network indeed separates the 4 groups into separate components and also captures a pattern specific to Group a: high relative abundances of both *Alistipes A* and *Marvinbryantia* are associated with a positive phenotype for samples in Group a. Although species *Marvinbryantia sp900066075* was the true predictor in our simulation, the genus *Marvinbryantia* was instead selected by the predictive model—likely due to the high redundancy among these closely related features. This example illustrates how endoR can depend on the fitted model. If endoR is fitted on a decision ensemble directly obtained from the true mechanism generating, rather than by fitting a predictive model, it indeed recovers the ground truth one (S2(A)–S2(C) Fig).

For a comparison with a null model, we de-correlated the artificial phenotype from the relative abundances by fully randomizing the target within each group in order to conserve the same group (or covariate) structure. The RF classifier we obtained on these null data had an expected accuracy of 55.54±1.73, and 44 features were selected. After regularization on 100 bootstraps, endoR returned a unique stable decision corresponding to the imbalance in group c that had a high error of 0.48. Therefore, endoR could effectively eliminate all noise from the RF and did not return any false positives.

The same procedure was followed to generate global null models for 10 independent artificial phenotypes (see the next section); on average, the RF classifiers fitted on the global null models had an accuracy of 59.36±5.03. endoR did not find any stable decision for 6 of the global null models and only 1 stable decision for 3 models, which were all due to target imbalance in the groups (S3 Fig). In only one case did endoR find a stable decision ensemble made of 9 decisions; the interpreted RF of this global null model had an average accuracy of 67.9 ±2.49, which was the highest across models. These results confirm the ability of endoR to distinguish noise from truth.

## Evaluation of endoR on simulated data

In this section, we summarize our findings from evaluating endoR on multiple simulated datasets; further details and additional evaluations can be found in the S3 Text. The evaluation is based on two simulation configurations. The first configuration, referred to as *fully simulated data* (FSD). Unlike our previous simulation used for demonstrating endoR, the FSD were constructed by simulating both the features and response variable. Features are independent from each other, normally distributed, and all predictive associations are known (illustrated in S4 Fig). The second configuration, referred to as *artificial phenotypes* (AP), is similar to the simulation used to demonstrate endoR, in that AP simulations also comprise features from published human gut metagenomes comprising 2147 samples [40] and a response variable constructed from combinations of relative abundances of randomly chosen taxa (Fig 2). Hence, predictive variables are dependent and not all predictive associations are known. A more detailed description of the data is given in the Methods section.

**endoR is robust to changes in hyperparameters.** We generated 100 FSD and 50 AP datasets, processed them with varying endoR hyperparameters, and evaluated how these changes affected the ability of endoR to recover correct edges in the decision networks (Fig 3A and 3D, S5(C), S5(F) and S5(G)–S5(J) Fig).

First, we explored the effect of $\alpha$, which by construction is supposed to control the expected number of wrong decisions selected by endoR after bootstrapping. Accordingly, the number of TP and FP edges identified by endoR also increased with increasing $\alpha$ (Fig 3A and 3B). Even for small values of $\alpha$, endoR recovered many TPs while controlling for low numbers of FPs. Furthermore, regardless of $\alpha$, TP edges were attributed the highest importances, hence resulting in high weighted precision of final decision ensembles (S1 Table).

Second, we varied the number of bootstraps on which our stability selection procedure is applied. Varying the number of bootstrap resamples between 10, 50, and 100 for FSDs, and 10 and 90 for APs, slightly increased the precision and sensitivity of endoR (Fig 3D and S5(F) Fig), and higher bootstrap numbers decreased the overfitting of results (S6 Fig). Generally a higher value for $B$ is preferable but the size is limited by computing resources (see S7(E) and S7(F) Fig for an evaluation of computing resources required). Our empirical results suggest that a value between 10 and 100 is often sufficient.

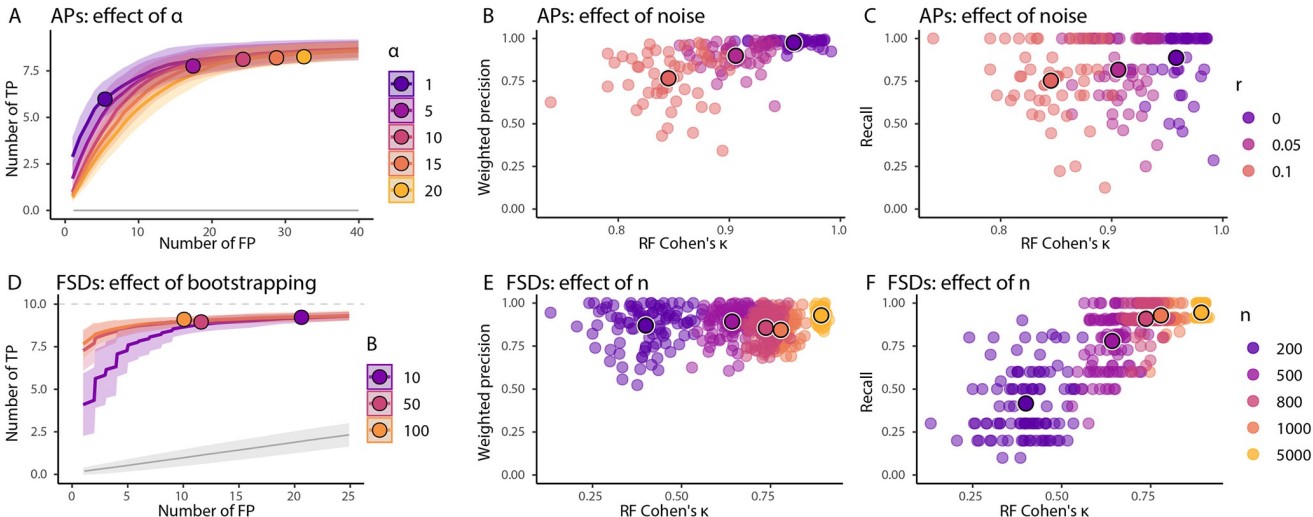

**Fig 3. endoR's performance is robust to hyperparameters and depends on the input model.** Simulation results based on 100 FSDs with $n = 1000$ observations (except when varied in E-F) and 50 APs using all observations (A-C). In all experiments, the noise was $r = 0.05$ (except when varied in B-C) and endoR was applied to fitted RFs with $\alpha = 5$ (except when varied in A) and $B = 10$ (except when varied in D). For each dataset and parameter setting we fitted a RF and applied endoR. Then, we computed the following three metrics: Cohen's $\kappa$ of the RF, weighted precision and recall values of the selected edges in the stable decision ensemble, and TP/FP-curves based on the probabilities of being selected in the stable decision ensemble (see Methods). A and D: TP/FP-curves are averaged across all datasets for a fixed parameter setting (line) and standard deviation (shaded area) are displayed. The average number of TPs and FPs expected for a randomization null model and standard deviations, are shown in grey. Large points indicate the average number of TPs and FPs in the stable ensembles generated by endoR. B-C and E-F: Each point corresponds to the precision/recall of endoR applied to a single dataset and parameter setting. The larger traced points are the averages across all datasets for a fixed parameter setting. A: Increasing $\alpha$ increases both the TPs and FPs. Small values of $\alpha$ effectively control the FPs without strongly impacting the recovered TPs. D: Larger values of $B$ are slightly better but endoR performs well even for small values of $B$. B-C and E-F: As expected decreasing the noise or increasing the number of observations improves the performance of endoR both in terms of precision and recall. Importantly, there is a strong dependence of endoR performance on the performance of the fitted RF. Moreover, endoR has a good precision even for small sample sizes.

Lastly, we assessed whether endoR's performance was affected by the method used for discretizing numeric value (e.g., grouping into 'Low' and 'High' numeric values; S8 Fig). endoR is indeed robust to the discretization procedure (S3 Text and S5(G)–S5(J) Fig).

**endoR improves with the accuracy of fitted models.** Since endoR interprets tree ensemble models, we evaluated the influence of the accuracy of RF models on the accuracy of endoR. We fitted RFs to 100 FSD and 50 AP datasets and applied endoR to the models. Model accuracy was altered by varying (i) noise levels via the $r$ parameter (Fig 3B and 3C and S5(A) and S5(B) Fig), (ii) the number of samples used to fit models (Fig 3E and 3F), and (iii) the model complexity through the number of trees in the forest (S5(D) and S5(E) Fig). Model accuracy increased with higher numbers of trees, lower noise, or higher number of samples (Fig 3B, 3C, 3E and 3F, S5(A), S5(B), S5(D) and S5(E) Fig).

On average, the weighted precision of endoR was high, even for low predictive model performances (i.e., small Cohen's $\kappa$; Fig 3E), and it increased with RF model performance as noise in data declined (Fig 3B). Importantly, even for small sample size (e.g., $n = 200$) endoR had high weighted precision values (Fig 3E). We attribute this to the regularization and resampling steps used in endoR, that effectively reduce the risk of overfitting. The endoR recall always increased with higher RF predictive performance (Fig 3C and 3F). Furthermore, the variance of the recall across datasets decreased with increased predictive performances, meaning that although endoR produces precise networks, the probability of recovering as many true interactions as possible increases with the predictive model accuracy. Taken together, the results

consistently showed that the performance of endoR depends on the quality of the input model (Fig 3B and 3E).

**endoR outperforms state-of-the-art methods for metagenome data analysis.** We utilized 50 AP datasets to evaluate the performance of endoR relative to the state-of-the-art (Fig 4 and S2 Text; results for the 100 FSD are provided in S9 Fig). Our evaluations included non-parametric statistical Wilcoxon rank-sum and $\chi^2$ tests, sparse covariance matrices computed with the sparCC [47] and graphical lasso (gLASSOciteFriedman2008glasso) methods, the Gini importance [19, 25], and SHAP values [28]. In particular, we used RFs to extract Gini importances, SHAP values of single variables, and endoR feature and interaction importances. Given that SHAP interaction values are not readily available for RF models in R, we fitted gradient boosted models using the xgboost R-package [46] to extract SHAP values and interaction values, Gini importances, and endoR feature and interaction importances. Wilcoxon rank-sum and $\chi^2$ tests identified single variables significantly associated with the artificial phenotypes, while sparse covariance matrices discriminated pairs of variables significantly correlated in one but not the other phenotypic group.

Each of the 50 APs simulated from real metagenomes was processed with all methods. Single variables and pairs of variables were ranked by the output parameters of each method and compared with the ground truth network to build TP/FP curves (S2 Text). Each curve displays the number of TP variables, or interaction effects between pairs of variables, found by each method for a given number of FP on average across the 50 APs (Fig 4).

All methods that did not use a predictive model (i.e., non-parametric statistical tests, sparCC, and gLASSO), performed poorly, with accuracies nearly equivalent to random guessing (Fig 4). Overall, single variables were very well ranked by endoR, SHAP, and Gini importances, with close to all TP attributable to the highest importances before any FP (Fig 4A).

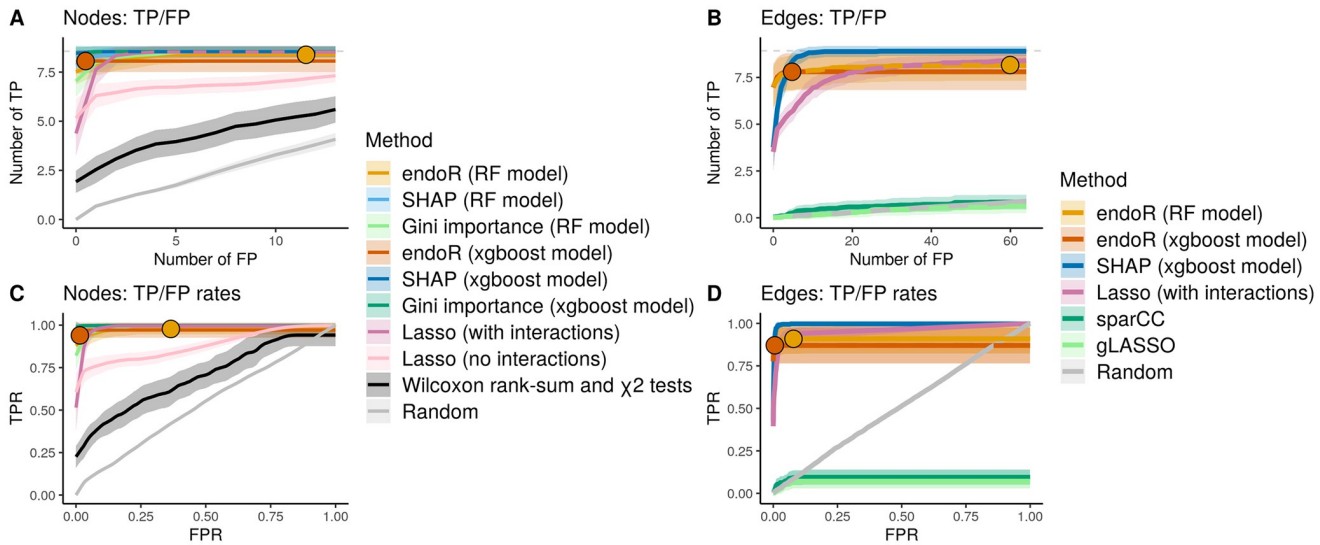

**Fig 4. endoR is better or comparable to state-of-the-art methods at identifying true variables and pairs of variables predictive of artificial phenotypes.** Average (line) and standard deviation (area) of identified true positive (TP) for a given number of false positive (FP). The average numbers of TP and FP in the endoR final decision ensemble are indicated with points. A, C: correspond to single variables and B, D: to pairs of variables across 50 replicates of artificial phenotypes. A, B: the truncated lines of absolute numbers of TP and FP are displayed, dashed grey lines denote the ground truth number of TP. C, D: the full curves of TP and FP rates are displayed. Lines are dashed when necessary due to overlaps. 'Random' signifies results expected with a randomization null model. A, C: All methods based on fitted predictive models almost perfectly ranked TP because of the feature selection step in model fitting. B, D: endoR better discriminated TP from FP edges than SHAP and lasso. Only endoR does not return all features and interactions, hence limiting the number of FPs in the final decision ensembles, although resulting in lower recall too.

SHAP and Gini had a better recall, but endoR was the only method to return a subset of variables, hence limiting the number of FP.

Interactions were identified with a higher recall by endoR from RF than XGBoost models in these simulations (Fig 4B), even though the average Cohen's $\kappa$ of XGBoost models was higher compared to RFs (on average across the 50 repetitions, from the mean across 10 CV sets for each replicate, Cohen's $\kappa = 0.97\pm0.00$ and $0.91\pm0.03$ for XGBoost and RF models, respectively). endoR was more accurate than SHAP at ranking of interactions. Again, SHAP recall was higher but the number of FP was limited in the endoR decision ensemble due to the selection of variables via regularization. Furthermore, endoR could extract interaction importances from RF models, while SHAP is not available in R for this purpose (Fig 4B). Hence, endoR outperformed other methods in terms of accuracy of results.

We note that the summary plots generated by endoR, specifically the feature importance and influence plot and the decision network, enable rapid assessment of important variables and the direction of their association with the response variable, as well as interactions between variables. On the contrary, SHAP values are designed to inform at the per-observation level and are not suited to provide general overview of results, especially as $p$ increases (S10 and S11 Figs). Therefore, in terms of data interpretability, endoR better suits analyses of metagenomes than SHAP, given that (i) $p$ is usually high, and (ii) many variable interactions are expected.

We compared SHAP and endoR in regards to computational performance. The two methods were compared on RF models only, since SHAP values are computed by the xgboost R-package [46] while fitting the model instead of post-hoc. SHAP values were generated from RF using the shap function from the iBreakDown R-package [48]. We found endoR to be substantially faster than shap. Specifically, endoR scales linearly with dataset dimensionality and sample size, while shap scales superlinearly (S7(A) and S7(C) Fig). As expected endoR CPU usage scaled linearly, increasing with the number of bootstraps (S7(E) Fig). We note that since endoR can be trivially parallelized, endoR requires less wall-time with either $B = 10$ or 25 than shap for the same number of threads (S7(G) Fig). endoR requires more memory than the shap function, but both scale sublinearly with dataset dimensionality and require only a few gigabytes for the maximum of 100 features and 2000 observations used for the evaluations (S7 (B), S7(D) and S7(F) Fig).

In summary, based on our evaluations on all simulated datasets and phenotypes, endoR performance is comparable or better than state-of-the-art methods in regards to identifying variables and interactions of variables associated with a response variable, while generating results that are easier to interpret in shorter computation times. Altogether, endoR surpasses state-of-the-art methods for analysing metagenome data.

## endoR rediscovers previously reported associations between cirrhosis and gut microbial composition

To illustrate the utility of endoR for microbiome studies, we applied our proposed workflow including endoR (Fig 1) to a previously published gut microbiome dataset comprising patients diagnosed with cirrhosis versus healthy individuals [41]. The dataset included 130 Chinese subjects, among which 48% were healthy, 35% were women, with ages varying from 18 to 78 years old (mean = 45), and BMI ranging from 16 to 29 kg.m$^{-2}$ (mean = 22). Our full model consisted of an RF with feature selection (see Methods; Cirrhosis metagenomes). The model was fitted to predict the disease status (i.e., 'healthy' or 'cirrhosis'), of individuals based on their age, gender, BMI, and relative abundances of gut microorganisms derived from metagenomes (S2 Text). On CV sets, it had an average Cohen's $\kappa$ of $0.73\pm0.08$ and accuracy of $0.87 \pm0.04$.

endoR identified 25 stable decisions that used 20 features (S2 Text). Many taxa used in the stable network generated by endoR were taxonomically closely related to taxa identified in the original study, with 'closely related' defined as the immediate parent or child taxonomic classification in the taxonomy hierarchy (e.g., the *Bacteroides* genus is the child of the *Bacteroidaceae* family, while *Bacteroidaceae* is the parent of *Bacteroides*) (Fig 5 and S12(A) Fig). Namely, *Veillonella parvula* and *Streptococcus* were confirmed as being enriched in individuals with cirrhosis (Fig 4, S12(C) and S12(D) Fig), as observed by studies on different cohorts [12, 49]. Moreover, while the *Megasphaera* genus was significantly enriched in cirrhotic individuals in the original study, endoR further identified that the species *Megasphaera micronuciformis* was the most important one to discriminate gut microbiomes of healthy individuals from those of cirrhotic individuals (Fig 4 and S12(B) Fig). The species was detected in 24% of healthy individuals versus 85% of individuals with cirrhosis. Additionally, for the samples in which *Megasphaera micronuciformis* was detected, the average abundance was 10 times lower in healthy individuals compared to cirrhotic ones (respectively, $0.40 \pm 1.50 \cdot 10^{-4}$ and $4.26 \pm 10.41 \cdot 10^{-4}$). Intriguingly, neither the genus nor the species were identified in other cohorts [12, 49]. Thus, *M. micronuciformis* may be a marker of cirrhosis specific to the Chinese cohort sampled by Qin et al. [41]. We note that *M. micronuciformis* was originally isolated from a liver abscess and pus sample [50].

Certain associations identified in the original study were not detected by endoR (S12(A) Fig). This can be partially explained by the stringent feature selection step in our model construction, which reduced the feature space from 922 to 81 taxa. For instance, significantly lower relative abundances of *Alistipes* (family *Rikenellaceae*) were found in individuals with cirrhosis by [41] and in other cohorts [12, 49]. In our analysis, relative abundances of *Alistipes* were not used by the model to classify diseased and healthy samples (Fig 4), which is likely due to the large overlap in *Alistipes* relative abundance distributions between healthy and cirrhotic individuals (S12(F) Fig). However, *Rikenella microfusus*, the other genus of the *Rikenellaceae* family detected in the dataset, showed lower overlap in relative abundances between cirrhosis (depleted) and healthy individuals (enriched); thus, it was selected and used by the model (Fig 4 and S12(F) Fig). In another example, the *Pasteuralleceae* family was found to be enriched in cirrhotic individuals with endoR but not in the original study (Fig 5, S12(A) and S12(G) Fig). However, the two most abundant genera in the *Pasteurellaceae* family, the *Haemophilus* and *Aggregatibacter* genera, were identified as differently enriched in healthy versus cirrhosis individuals by Qin et al. [41] (Fig 4, S12(A) and S12(H) Fig). In conclusion, some of the discrepancies between our analysis and the original study may be due to our use of a RF model, which can integrate non-linear associations. Also, our feature selection step selected taxa often closely related to the genera identified in the original study, indicating that these sister taxa are actually more predictive of cirrhosis when using a RF model.

endoR identified new associations between cirrhosis and the gut microbiome. Among others, we found additional oral-microbiome associated taxa to be enriched in cirrhotic individuals. For instance, endoR revealed an important enrichment in the *Leptotrichia* genus in individuals with cirrhosis (Fig 5A). This taxon is part of the oral microbiome [51] and is enriched in patients with periodontal disease [52]. In addition, endoR identified an enrichment of the oral-taxon *Kingella denitrificans*, a member of *Neisseriaceae* [51], in individuals with cirrhosis (Fig 5A). Altogether, these findings support the hypothesis of Qin et al. [41], in which oral commensals colonized the guts of patients with liver cirrhosis.

Our analysis also revealed an important depletion of *Adlercreutzia equolifaciens*, a bacterium associated with healthy individuals [42] (Fig 5A and S12(E) Fig). Additionally, the decision network extracted from the stable decision ensemble contained only a few edges (Fig 5B),

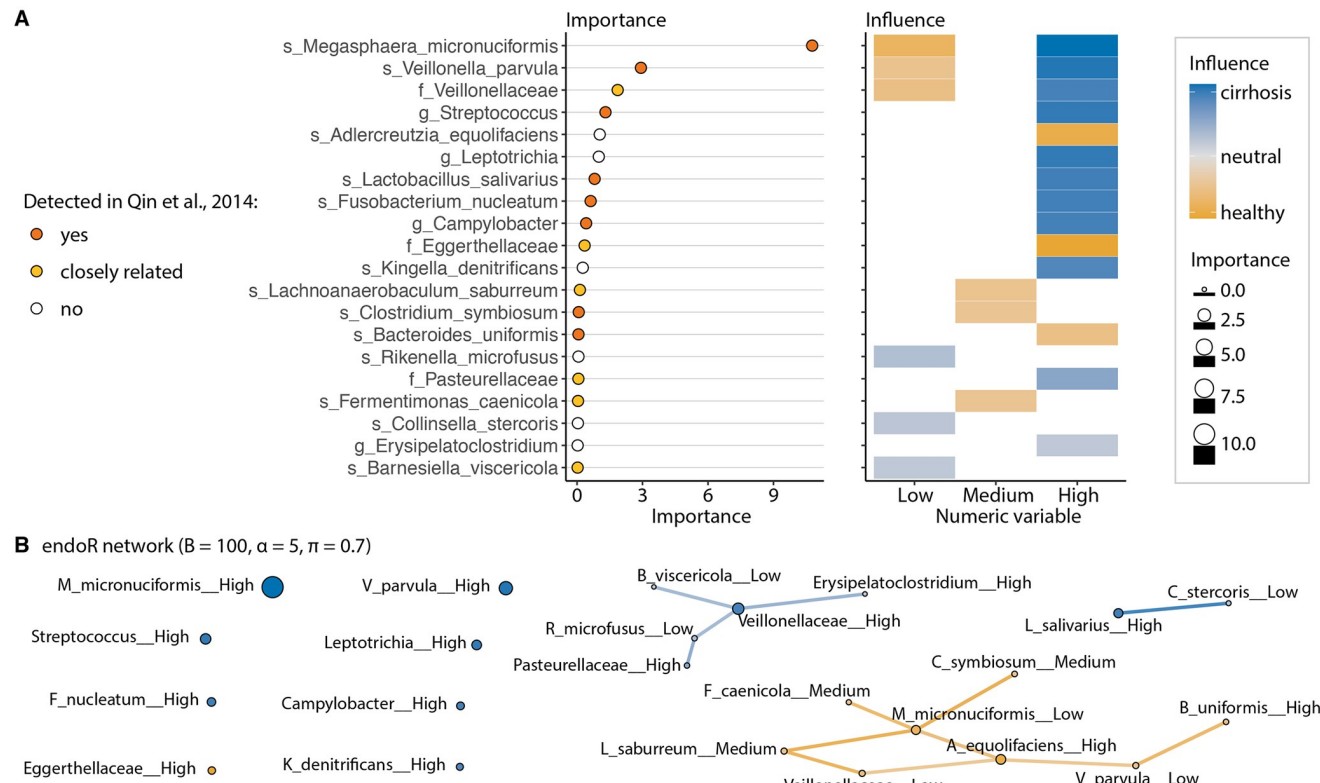

**Fig 5. endoR recapitulates previous findings on differences in gut microbiomes between healthy individuals and patients diagnosed with cirrhosis.** A: Feature importance aggregated across each level of discretized variables and influence per-level as determined by endoR. Levels correspond to discrete variable categories, and here represent relative abundance groups created by endoR (i.e., whether samples had 'Low', 'Medium' or 'High' relative abundances of each taxon). 'closely related' designates taxa that are the direct parent or child taxonomic classification of a taxon originally associated with disease status in Qin et al. [41]. White boxes in the influence plot signify that the level was not used in any stable decision; thus, the influence could not be calculated. B: Decision network extracted from the stable decision ensembles. See Fig 2 for the description of the network; the boxed legend is shared for A and B.

meaning that a few interaction effects of bacteria on cirrhosis were found. Hence, our analysis suggests few higher order interactions among gut microbiota in relation to cirrhosis.

Given the few interactions extracted by endoR from the RF model, we posited that a simpler linear regression model may perform as well as our more complex random forest. Accordingly, we fitted two linear regression models with lasso penalty [53] to predict the healthy or cirrhotic state of individuals: the first model only included main effects, while the second additionally included all pairwise feature interactions. The model without interactions had a better accuracy than the model including all pairwise interactions (respectively, 0.81±0.07 and 0.79±0.08, averaged across 10 CV sets), indicating that considering all pairwise interactions added more noise than information to the model. However, both lasso models were less accurate than the RF model, which had an average accuracy of 0.87±0.04 across 10 CV sets. These findings support the relevance of interactions identified by endoR as improvements to the model.

### New insights into the ecology of human gut *Methanobacteriaceae*

We utilized endoR to gain insight into the factors influencing the prevalence of *Methanobacteriaceae* in the human gut. We focused on this microbial clade because (i) *Methanobacteriaceae* are the most prevalent and abundant archaea in the human gut [54, 55], (ii)

methanogenic archaea influence bacterial fermentation via $H_2$ consumption [56–58], (iii) species of *Methanobacteriaceae* have been shown to form a complex trophic network with certain bacteria [2, 55, 58–64], and (iv) *Methanobacteriaceae* have been associated with various host phenotypes such as constipation and slow transit [65, 66], non-western diet [67–69], and body mass index (BMI) [58, 59, 70–78]. Therefore, *Methanobacteriaceae* is a prime candidate for the application of endoR to resolve how this clade associates with bacterial taxa and host factors (e.g., BMI).

Metagenomes gathered for this analysis comprised 2203 individuals from 26 studies living in 23 countries across the globe (S2 and S3 Tables). Participants varied in ages from 19 to 84 years old, with a median and mean age of 33 and 40 years old, respectively. BMI ranged from 16.02 to 36.41 kg.m$^{-2}$, with median and mean values of 23.27 and 24.03 kg.m$^{-2}$. Women comprised 62.30% of individuals, and 76.53% of individuals were from westernized populations [79, 80].

We trained an RF with feature selection (see Methods; *Methanobacteriaceae*) to predict the presence of *Methanobacteriaceae* in the human gut by using taxon and metabolic pathway relative abundances, host descriptors, and metadata (see the S2 and S3 Tables, for a description of samples, host descriptors and metadata included, and S2 Text and S1 Fig, for a description of model selection and fitting). Metadata comprised the number of reads and dataset names, and both were always included for feature selection and fitting of the classifier to make sure that algorithms could correct for these variables, if necessary (e.g., in case of batch effects) [81]. To evaluate the association between the presence of *Methanobacteriaceae* and host descriptors with incomplete information across samples (no age, BMI, and gender information was reported for 528, 1183, and 432 individuals, respectively), we subset observations to the 748 samples with complete information and applied our model fitting procedure. The best performing model had an average accuracy of 0.80±0.03 and Cohen's $\kappa$ of 0.55±0.06, based on unseen observations (S4 Table). Age, BMI, and gender were never selected across any CV set of this model. Therefore, they were excluded from further analyses and only human descriptors with complete information were included in the set of variables used to select the final model on all 2203 observations, such as country of sampling (S3 Table).

The final model accuracy and Cohen's $\kappa$ was 0.82±0.01 and 0.60±0.03, respectively (S2 Text and S4 Table). For the purpose of data interpretation via endoR, we trained a model on all observations and included 107 features selected by the taxa-aware gRRF algorithm as well as the metadata (S13(B) Fig).

A stable decision ensemble was extracted from the predictive model using endoR with $\alpha = 5$ and 100 bootstrap resamples. The ensemble comprised 60 decisions that could make predictions on all samples, with an average decision error of 0.40±0.07 and support of 0.37±0.12 (S6 Table). A total of 34 features were used in decisions to predict the presence of *Methanobacteriaceae* (Fig 6A and S7 Table). Feature importances were consistent between endoR and the mean decrease in Gini index (S13 Fig).

Both the *CAG-138* family (order *Christensenellales*, class *Clostridia*) and the *Phil-1* genus within *CAG-138* had the highest feature importances (Fig 6A). The *Oscillospirales* order (class *Clostridia*) was over-represented in features used by endoR compared to what would be expected by random ($p$-value = $10^{-3}$, S8 Table), with 15 taxa from this order of the 272 taxonomic features (family, genera, and species) detected in the dataset and included in decisions (Fig 6A). The *RF39* order (class *Bacilli*) was also over-represented ($p$-value = $10^{-3}$, S8 Table). Most taxa belonged to the *Clostridia* class (26 taxa, S8 Table) and had relatively higher importances compared to other features. Accordingly, the relative abundance of the *Clostridia* class was significantly associated with the presence of *Methanobacteriaceae* (Wilcoxon rank-sum test, $p$-value = $1.18 \cdot 10^{-20}$, S14 Fig).

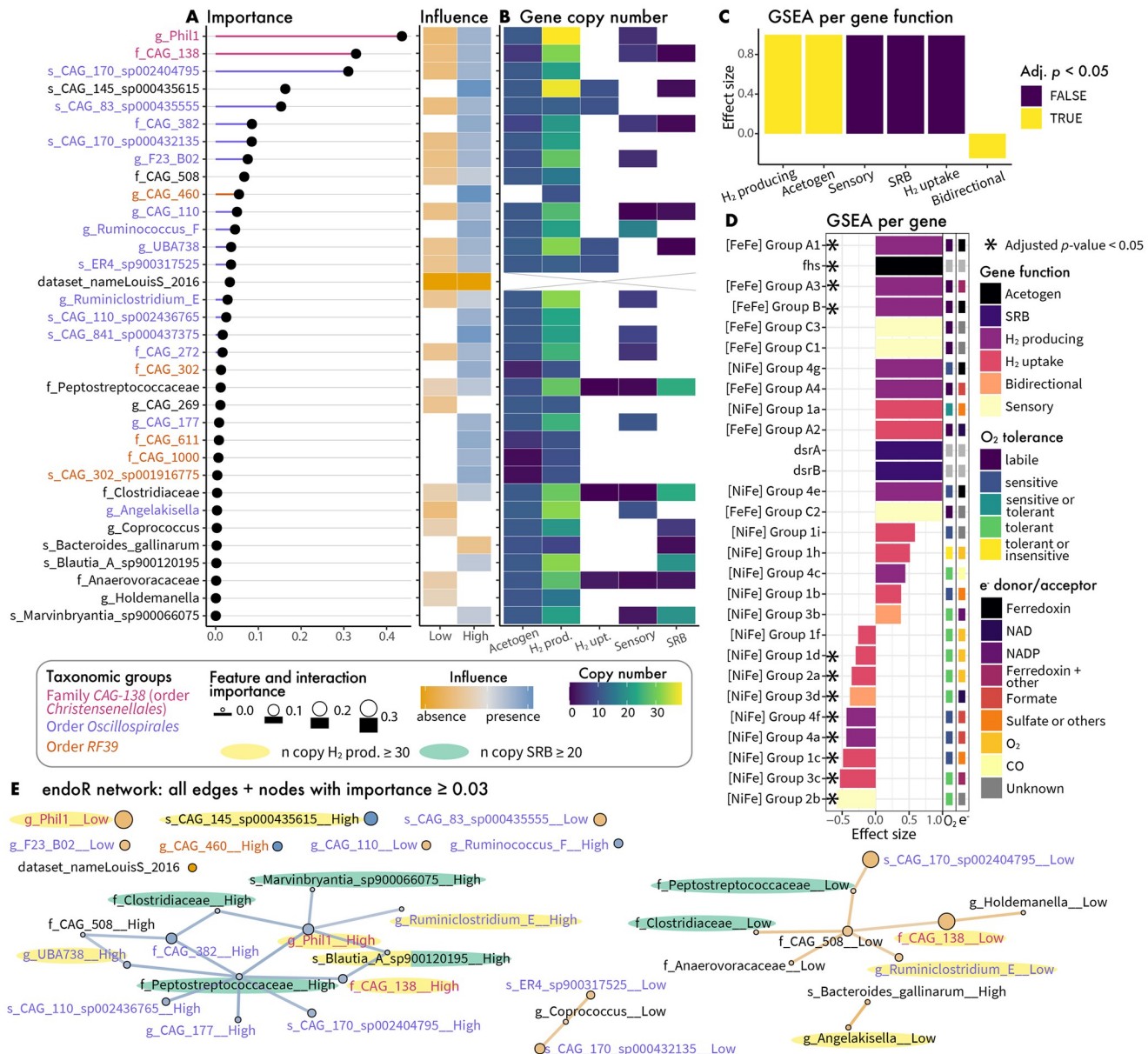

**Fig 6. Relative abundances (RA) of *Oscillospirales*, *Christensenellales* and other select bacteria predict conditions favorable to colonization of the human gut by *Methanobacteriaceae*.** A: Feature importance and influence for each taxa used by the decision ensemble generated by endoR. Taxonomic levels are indicated with label prefixes: 'f_' = family, 'g_' = genus, and 's_' = species, while taxonomic orders are indicated via bar and label colors. Levels correspond to 'Low' and 'High' relative abundances of taxa. B: Sum of gene copy numbers of marker genes involved in $H_2$ production and consumption (see Methods and S5 Table), for endoR selected features. SRB: *dsrA* and *dsrB* genes exclusively involved in sulfate reduction [82]; Acetogen: *fhs* gene involved in acetogenesis [83]; other categories correspond to hydrogenases predicted functions as determined by the HydDB database: $H_2$ production ($H_2$-prod.), $H_2$ uptake ($H_2$-upt.), sensory [84]. Boxes are white for taxa for which genes were not detected in their genomes. The cross indicates 'Non applicable' (for the 'dataset_nameLouisS_2016' feature). C-D: Effect sizes from gene set enrichment analyses performed at the gene function (C/) or for each gene (D/), bars are colored by the adjusted *p*-values (Adj. p). D: Bars are colored by gene function. The predicted $O_2$ tolerance of hydrogenases and electron ($e^-$) donor or acceptor are indicated by colored boxes on the right of the plot [84]. Asterisks denote significance (adjusted *p*-value < 0.05). E: Decision network in which nodes correspond to individual features and edges correspond to pairwise interactions. Nodes and edges colors describe the feature and interaction influence; their sizes and widths are proportional to their importances. Nodes with an importance ≥ than 0.3 but not connected are shown. Taxa with a gene copy number ≥ 30 for $H_2$ production and ≥ 20 for SRB genes are highlighted in yellow and green, respectively. The boxed legend applies to A, B, and E.

We note that none of the host descriptors or metabolic pathways were predictive, indicating that microbiome taxonomic composition may be more important for determining the prevalence of *Methanobacteriaceae*. However, we must acknowledge that (i) host descriptors were limited, and (ii) metabolic pathway diversity was likely undersampled. Interestingly, the model identified a cofounding effect due to possible dataset bias: samples from the LouisS_2016 study were indeed depleted in *Methanobacteriaceae*. This dataset comprised 92 stool samples from German individuals, of which *Methanobacteriaceae* was never detected. The authors utilized a non-standard DNA extraction protocol [85], which may explain the lack of *Methanobacteriaceae* detection, given that extraction protocols differ substantially in their lysis efficiency of methanogenic archaea [86, 87].

To assess whether bacterial taxa selected by endoR may be part of a $H_2$-based syntrophic network, we estimated the number of genes involved in $H_2$ production and consumption for the 33 taxon features (Fig 6B). Specifically, we utilized representative genomes and assessed (i) genes coding for hydrogenases involved in $H_2$ production, $H_2$ consumption, both (bidirectional), or $H_2$ sensing [84], (ii) genes involved exclusively in sulfate reduction (*dsrA* and *dsrB*) [82], and (iii) genes involved in acetogenesis (*fhs*) [83] (Fig 6B and S15 Fig). To determine which of these genes were enriched among the endoR-selected features, we conducted a gene set enrichment analysis [88] based on endoR importance values. When we grouped genes by function (e.g., '$H_2$ uptake' or 'SRB'), $H_2$-production and acetogens were significantly enriched, while bidirectional hydrogenases were depleted (adjusted $p$-value $< 10^{-3}$, Fig 6C). In particular, 22 of the 33 taxa possessed more than 20 copies of genes coding for hydrogenases involved in $H_2$-production (Fig 6B). At the per-gene level, the acetogen marker gene (fhs), along with the $H_2$-producing [FeFe] Group A1, A3, and B hydrogenases were significantly enriched, while many [NiFe] hydrogenases were significantly depleted (adjusted $p$-values $< 10^{-2}$, Fig 6E). Furthermore, there was a clear gradient of higher $O_2$ sensitivity for enriched hydrogenases and increased $O_2$ tolerance for depleted hydrogenases (Fig 6D). These results suggest that *Methanobacteriaceae* co-occurs with acetogens and $H_2$-producing bacteria possessing [FeFe] hydrogenases, while the negative association between bacteria possessing $O_2$-tolerant [NiFe] hydrogenases suggests $O_2$ exposure may be a common cause of *Methanobacteriaceae* absence.

Interestingly, the endoR decision network showed a strong positive association between *Phil-1* and the four taxa with the highest number of *dsrA* and *dsrB* gene copies: *Clostridiaceae*, *Peptostreptococcaceae*, *Blautia A sp900120195*, and *Marvinbryantia sp900066075* (Fig 6B and 6E). The influence of these $H_2$-consumers is not pronounced, but the interaction effect between the relative abundances of these taxa and the high relative abundances of *Phil-1* is clearly associated with the presence of *Methanobacteriaceae* (Fig 6E). While sulfate reducers generally out-compete methanogens for $H_2$ [89, 90], *Phil-1* may generate enough to alleviate $H_2$ competition. Alternatively, the sulfate reducers or *Methanobacteriaceae* may be utilizing alternative substrates for growth.

## Discussion

Applying machine learning to microbiome data has increased in popularity due to the approach's compatibility with the high-dimensional, compositional, and zero-inflated properties of amplicon and shotgun metagenome data [15, 16, 18]. However, interpreting machine learning models to gain mechanistic insight into processes underpinning microbial diversity and ecosystem functioning can be challenging. We showed, through extensive validation on simulated and real microbiome data, that our proposed procedure endoR addresses these difficulties by recovering and visualizing the important components of tree based machine learning models. First, the accuracy of identifying important features and the interactions among

features surpassed or at least rivaled existing state-of-the-art methods (Fig 4). Second, the feature importance and influence plots and decision networks generated by endoR were straightforward to interpret and provided more information than existing methods (Fig 2H–2J versus Fig 2G, S5 and S9 Figs). Third, endoR was robust to the choice of hyperparameters (Fig 3A and 3D) and, by including several regularization steps (e.g., resampling inspired by stability selection [91]), effectively controlled false discoveries, even in settings with small sample sizes (Fig 3E). Fourth, endoR is flexible: it can be applied to both random forests and gradient boosted trees, which themselves can be applied to both regression and classification tasks involving various types of features (e.g., microbial abundances and metadata). Finally, endoR is substantially more computationally efficient than, for example, SHAP (S7(A)–S7(D) and S7 (G) Fig), which is a common approach for ML model interpretation [31, 35, 92, 93].

Our re-evaluation of healthy and cirrhotic individuals initially assessed by Qin et al. [41] highlights the ability of endoR to detect known microbe-disease associations while also revealing how microbial features interact in regards to disease status (Fig 5). For example, the feature importance calculated by endoR highlighted the main microbial factors previously shown to distinguish cirrhotic and healthy individuals—particularly emphasizing the importance of *M. micronuciformis* and *V. parvula*. Notably, our approach revealed microbe-disease associations not identified in the original study. endoR found additional bacteria common in the oral microbiome to be enriched in gut microbiome of cirrhotic individuals, among which one was associated with periodontitis [52], a condition more prevalent in individuals with alcohol-related cirrhosis, presumably due to a decrease in oral hygiene [94]. endoR also found *Adlercreutzia equolifaciens* to be depleted in individuals with cirrhosis (Fig 5A). This bacterium is associated with healthy individuals compared to ones suffering from primary sclerosing cholangitis, which can lead to cirrhosis [42].

Given the importance of methanogens for mediating bacterial fermentation via syntrophic $H_2$ exchange, we applied endoR to understand which bacteria and host factors determine the presence of *Methanobacteriaceae*, the dominant methanogenic clade in the human gut. Our extensive dataset, comprising a global collection of 2203 samples from 26 studies, allowed for a robust assessment across disparate human populations. endoR identified 33 bacterial clades to be predictive of *Methanobacteriaceae*'s presence. In particular, we confirmed the strong association previously observed between *Methanobacteriaceae* and members of the *Christensenellales* order [58–62], particularly with the uncultured *CAG-138* family (Fig 6A). We also found members of the order *RF39* (class *Bacilli*) to be positively associated with *Methanobacteriaceae*. This is consistent with findings from [59] who described that *RF39* and *Methanobacteriaceae* belong to a consortium of co-occurring taxa, with *Christensenellales* forming the central hub. *RF39* are uncultivated microorganisms with very small genomes and are predicted to be acetogens [95, 96]. Hence, the co-occurrence of *RF39* and *Methanobacteriaceae* may be a result of their affinity for $H_2$ produced by *Christensenellales*. Nonetheless, contrary to other $H_2$-consumers, no interaction effect was found between members of the *RF39* order and *Christensenellales* for predicting the presence of *Methanobacteriaceae* (Fig 6E). As acetogenesis is a facultative metabolic pathway and *RF39* are predicted to produce $H_2$ [96], $H_2$ syntrophy may be an additional underlying mechanism of the association between members of the *RF39* order and *Methanobacteriaceae*.

Our findings highlight the importance of $H_2$ production and consumption for predicting the presence of *Methanobacteriaceae* (Fig 6). Clades known to include acetogens and SRB were among the taxa positively associated with *Methanobacteriaceae*, which would seem to indicate competition for $H_2$; nonetheless, all competitors were positively associated and seemingly can coexist (Fig 6A and 6B). High rates of $H_2$ production may mitigate this competition. Indeed, $H_2$-producing [FeFe] hydrogenases have very high turnover rates compared to [NiFe]

hydrogenases [97], and they were the only hydrogenases enriched among the endoR-selected bacteria (Fig 6D). Moreover, the enriched [FeFe] hydrogenases are $O_2$ labile [84], utilize the low redox electron carrier ferredoxin [98], and are associated with obligate anaerobes [99]. This contrasts the generally $O_2$-tolerant [NiFe] hydrogenases not utilizing ferredoxin and that were depleted among endoR-selected taxa [84]. These findings suggest that intestinal aerobiosis may mediate the presence of both *Methanobacteriaceae* and bacteria positively associated with the clade due to the low redox required for methanogenesis, along with the $O_2$ sensitivity of *Methanobacteriaceae* and the bacterial $H_2$ producers possessing [FeFe] hydrogenases. The absence of both *Methanobacteriaceae* and these $H_2$ producers may indicate epithelial oxygenation resulting from diseases such as IBD or ulcerative colitis [100–102]. Indeed, a decline in *Methanobacteriaceae* taxa has been associated with IBD, ulcerative colitis, and Crohn's disease [103–105].

Intestinal transit times may also be a factor determining *Methanobacteriaceae*'s prevalence. Many of the endoR-selected bacteria are members of the *Oscillospiraceae* family whose members are predicted to have slow replication times and would thus benefit from slow transit times [106]. Similarly, *Methanobacteriaceae* species have generally slow replication rates and are associated with increased transit time [107, 108]. Moreover, $CH_4$ can slow peristalsis [109], hence methanogenesis may be indirectly promoting the persistence of *Oscillospiraceae* species via manipulating host physiology.

Still, no host factors were predictive, including BMI, while previous work has shown associations between *Methanobacteriaceae* taxa (or methanogens assessed in aggregate) with either anorexic, lean or obese phenotypes, depending on the study [58, 59, 67–78]. These contradictory findings among existing studies, and our lack of association between BMI and *Methanobacteriaceae*, suggest that population-specific or study-specific factors mediate this association. While endoR could identify such context-dependent associations, our aggregated dataset may not contain the relevant factors (e.g., diet or other lifestyle factors). Westernization status was also not predictive of *Methanobacteriaceae*, although taxa in this clade have been found to be enriched in certain non-westernized populations such as Matses hunter-gatherers [67, 68], traditionally agricultural Tunapuco [67], or Columbians in the midst of westernization [110]. The categorization of 'westernized' versus 'non-westernized' is likely overly broad to accurately predict *Methanobacteriaceae* across disparate human populations (S15(H)–S15(K) Fig). Indeed, not all studies have shown enrichment of *Methanobacteriaceae* in 'non-westernized' populations [80, 111, 112].

In summary, endoR advances the state-of-the-art for interpreting machine learning models trained on microbiome amplicon and shotgun metagenome data. We note that regardless of the ML model interpretation method, poor model performance will generate misleading interpretations. Our evaluations of endoR's accuracy in regards to the tree ensemble model accuracy provide an explicit guideline for evaluating the trustworthiness of model interpretations generated by endoR. Moreover, we provide sensible parameter defaults that will often lead to robust results, but we emphasize the careful consideration of parameters based on our extensive evaluations. As we show with our validations and application to gut-inhabiting *Methanobacteriaceae*, endoR produces robust and informative model interpretations. These allow researchers to gain insight into the biological mechanisms underpinning ML model predictive performance and help guide controlled experimentation to directly test such mechanisms.

## Materials and methods

This section is divided into three parts: (i) a detailed description of endoR, (ii) a summary of the evaluation metrics, and (iii) an overview of the simulated and real data. Further methodological details and the technical implementation are covered in the S1 Text.

## Description of endoR

endoR takes a fitted tree-based ML prediction model and extracts a regularized decision ensemble. Based on this decision ensemble it computes decision, feature, and interaction importance and influence metrics by assessing their individual contribution to the overall prediction. These metrics are visualized in easy-to-interpret plots that can be used to gain insights into the fitted model. In the following, we describe the mathematical details underlying the decision ensembles and metrics, explain how endoR regularizes the decision ensemble, and present how results are visualized.

**Rules, decisions and decision ensembles.** Let $\mathbf{x} = (x^1, \ldots, x^p) \in \mathbb{R}^p$ represent $p$ features (numeric or factor variables, e.g., relative abundances of taxa and host gender), $y \in \mathbb{R}$ a response variable, and assume we have observed a sample of $n$ observations $(\mathbf{x}_1, y_1), \ldots, (\mathbf{x}_n, y_n) \in \mathbb{R}^{p+1}$. Our framework is able to handle both regression ($y$ continuous) and binary classification ($y$ binary); endoR transforms multi-class classification tasks to binary problems (one class versus all others).

A *rule* is a function $r : \mathbb{R}^p \to \{0, 1\}$ of the form

$$r(\mathbf{x}) = \mathbb{1}_{\mathcal{X}_r}(\mathbf{x}) = \prod_{j=1}^{p} \mathbb{1}_{\mathcal{X}_r^j}(x^j), \tag{1}$$

where $\mathcal{X}_r = \mathcal{X}_r^1 \times \cdots \times \mathcal{X}_r^p \subseteq \mathbb{R}^p$. We define a *decision* to be a tuple $D = \{r_D, \hat{y}_D\}$ consisting of a rule $r_D$ and a constant *prediction* $\hat{y}_D$. The prediction $\hat{y}_D$ is computed during model fitting following any pre-defined estimation procedure (e.g., least-squares) and should be thought of as a good approximation of $y$ on the *sample support* $S_D := \{i \in \{1, \ldots, n\} | r_D(\mathbf{x}_i) = 1\}$, the subset of samples following the rule. Decisions are the building blocks of a large class of non-parametric ML models such as random forests and boosted trees. These models combine many decisions to construct high-capacity prediction procedures. Any such model can be seen as a collection of decisions $\mathcal{D} = \{D_1, \ldots, D_M\}$, which we call a *decision ensemble*, together with an appropriate method for aggregating the predictions [20].

For every subset of observations $S \subseteq \{1, \ldots, n\}$, we define the error function $\alpha(S, \cdot) : \mathbb{R} \to \mathbb{R}$ either as the mean residual sum of squares in the case of regression, or by the mean misclassification error in the case of binary classification, formally,

$$\alpha(S, \hat{y}) := \frac{1}{|S|} \sum_{i \in S} (y_i - \hat{y})^2$$

or

$$\alpha(S, \hat{y}) := \frac{1}{|S|} \sum_{i \in S} \left(1 - (\hat{y})^{y_i}(1 - \hat{y})^{1-y_i}\right),$$

respectively. For a fixed decision $D$ and a variable $x^j$, or pair of variables $\{x^j, x^k\}$, we define the *complement decision* $D_j^c$, or $D_{j,k}^c$, to be the decision resulting from modifying the rule $r_D$ to have the complement support for the variable $x^j$, or for the pair of variables $\{x^j, x^k\}$. Additionally, we define the decisions $D_j^{rm}$ and $D_{j,k}^{rm}$ to be the decisions resulting from removing the variable $x^j$, or the pair of variables $\{x^j, x^k\}$, from the rule $r_D$. See the S1 Text and S16 Fig, for a visualization of these modified decisions. The predictions $\hat{y}_{D_j^c}, \hat{y}_{D_{j,k}^c}, \hat{y}_{D_j^{rm}}$ and $\hat{y}_{D_{j,k}^{rm}}$ are each updated based on the new rule.

For a variable $x^j$, we define the set of *active decisions* as $\mathcal{D}^j := \{D \in \mathcal{D} | \mathcal{X}_{r_D}^j \neq \mathbb{R}\}$, which is the subset of decisions which depend on $x^j$. Likewise, the set of active decisions of a pair of variables $\{x^j, x^k\}$ is defined as $\mathcal{D}^{j,k} := \mathcal{D}^j \cap \mathcal{D}^k$.

**Decision importance.** For a decision $D \in \mathcal{D}$, we define the *decision importance* by

$$I_D := \left(1 - \frac{\alpha(S_D, \hat{y}_D)}{\alpha(S_D, \bar{y})}\right) \cdot |S_D|.$$

This quantifies the improvement of predicting $y$ on the support $S_D$ with $\hat{y}_D$ instead of with the full sample average $\bar{y} := \frac{1}{n}\sum_{i=1}^{n} y_i$. It is weighted by the size of the decision's support.

For regression and binary classification, $\left(1 - \frac{\alpha(S_D, \hat{y}_D)}{\alpha(S_D, \bar{y})}\right)$ corresponds to the coefficient of determination (or $R^2$) [113] and Cohen's $\kappa$ [114], respectively, computed on the subsample $S_D$. Thus, the decision importance is a quality measure that incorporates both the support size and predictive performance of the decision.

**Feature and interaction importance.** For a variable $x^j$, we define the *decision-wise feature importance*

$$\delta_D^j := \alpha(S_D, \hat{y}_{D_j^{\mathrm{rm}}}) - \alpha(S_D, \hat{y}_D),$$

as the difference in predictive performance on $S_D$ between $\hat{y}_D$ and $\hat{y}_{D_j^{\mathrm{rm}}}$ (i.e., utilizing versus not utilizing information about $x^j$ for the prediction, S16 Fig).

For a pair of variables $\{x^j, x^k\}$, the *decision-wise interaction importance*

$$\delta_D^{j,k} := \sqrt{\delta_D^j \delta_D^k}$$

is the product of the decision-wise feature importances of $x^j$ and $x^k$ (see S1 Text for details on the rational behind this expression). We use the square root to ensure that the interaction importance remains on the same scale as the feature importance.

The *feature importance* and *interaction importance*,

$$F_j := \sum_{D \in \mathcal{D}} \delta_D^j I_D \qquad \text{and} \qquad F_{j,k} := \sum_{D \in \mathcal{D}} \delta_D^{j,k} I_D,$$

respectively, are then obtained by summing decision-wise feature and interaction importances over all decisions in $\mathcal{D}$ weighted by the decision importance. High values of the feature and interaction importances indicate that the variable, or pair of variables, contribute strongly to important decisions.

**Feature and interaction influence and direction.** For every decision $D$ and variable $x^j$, we define the *direction indicator* $d_D^j \in \{-1, 1\}$ to express whether a rule predominantly uses small or large values of that variable (see S1 Text). And, we calculate $\eta_{j,k} := \mathrm{sign}\left(\sum_{D \in \mathcal{D}^{j,k}} (d_D^j \cdot d_D^k \cdot I_D)\right)$ to record whether variables $\{x^j, x^k\}$ are each associated with $y$ in the same direction.

To measure the influence of a feature, or pair of features, on the prediction $\hat{y}_D$ of a decision, we proceed similarly as with the feature importance, though we now compare actual predictions instead of errors of predictions on $S_D$.

We define for a variable $x^j$ and a pair of variables $\{x^j, x^k\}$, the *decision-wise feature influence* and *decision-wise interaction influence* as

$$\gamma_D^j := d_D^j(\hat{y}_D - \hat{y}_{D_j^{\mathrm{rm}}}) \qquad \text{and} \qquad \gamma_D^{j,k} := \frac{d_D^j + d_D^k}{2}(\hat{y}_D - \hat{y}_{D_{j,k}^{\mathrm{rm}}}).$$

A large positive value of $\gamma_D^j$ indicates that large values of $x^j$ are positively associated with the response $y$ on the support of the rule, while negative values of $\gamma_D^j$ imply a negative association. Likewise, a large value of $\gamma_D^{j,k}$ indicates that large values of both $\{x^j, x^k\}$ are positively associated

with $y$, and $\gamma_D^{j,k}$ is negative when small values of both $\{x^j, x^k\}$ are negatively associated with $y$. In addition, $\gamma_D^{j,k}$ is equal to zero when the directions of association of the variables $x^j$ and $x^k$ with $y$ are opposite.

We assess the overall *feature influence* of a feature $x^j$, and *interaction influence* of pair of variables $\{x^j, x^k\}$, by averaging the decision-wise feature and interaction influences, respectively,

$$\Gamma_j := \frac{1}{\sum_{D \in \mathcal{D}^j} I_D} \sum_{D \in \mathcal{D}^j} \gamma_D^j I_D$$

and

$$\Gamma_{j,k} := \frac{1}{\sum_{D \in \mathcal{D}^{j,k}} I_D} \sum_{D \in \mathcal{D}^{j,k}} \gamma_D^{j,k} I_D$$

**Regularization of the decision ensemble.** We propose several procedures to regularize the decision ensemble and so reduce the noise by including a simplicity bias. Procedures are briefly introduced here but are presented in detail in the S1 Text.

**Decision-wise regularization.** The optional first and second steps involve discretization of numeric variables (into 2 categories by default) and pruning of rules. Pruning consists of removing variables from decisions that do not substantially participate in a decision (i.e., for which the difference in errors of the decision with and without the variable is low) [38]. After each decision-wise simplification step, decisions consisting of the same rules are grouped, the multiplicity is recorded (i.e., how many decisions have been collapsed into the simplified decision) and the prediction, error, support, and importances are re-computed based on the updated rule. Lastly, the decision importance is weighted by the decision multiplicity.

**Decision ensemble stability.** At this stage, the decision ensemble will often be large and still include poorly predictive decisions. In addition, the metrics (e.g., feature and interaction importances) may have a tendency to overfit. To avoid these issues, endoR implements an option to simplify the decision ensemble by running all decision-wise regularization steps and decisions metric calculations on $B$ bootstrap resamples of the data (this does not include refitting the prediction model). Bootstrapping is performed by sub-sampling with replacement. endoR then simplifies the decision ensemble by only keeping decisions that are returned consistently across bootstrap resamples. This approach is motivated by the stability selection procedure due to [91]. More specifically, for user-selected parameters $\alpha \in \mathbb{R}_{>0}$ and $\pi_{\mathrm{thr}} \in (0.5, 1]$ ($\pi_{\mathrm{thr}} = 0.7$ and $\alpha = 1$ by default), the $q$ most important decisions of each bootstrap resample are recorded and those appearing in at least $\pi_{\mathrm{thr}} \cdot B$ of the resampled decision ensembles are then selected. Motivated by the theoretical results of [91] on controlling the expected number of false discoveries ($\alpha$ corresponds to the expected number of false discoveries in this context), we select $q$ to be

$$q = \lfloor \max\{1, \sqrt{(2\pi_{\mathrm{thr}} - 1) \cdot \alpha \cdot d}\} \rfloor$$

where $d$ is the average number of decisions across all bootstrap resamples. For each decision in the stable decision ensemble, the decision-wise influence and importance are averaged across the resampled decision ensembles, and the influence and importance are re-computed as described above. By default, bootstrapping is performed on $B = 10$ resamples of size $n/2$.

**Visualization: Decision network and importance/influence plot.** After extacting a regularized decision emsemble and computing all metrics, endoR visualizes the results in a feature importance plot, a feature influence plot, and a decision network (summarized in Fig 1A and

exemplified in Fig 2H–2J). Both the feature importance and influence plots show only the main effects of variables that appear in the final regularized decision ensemble. For the influence plot, white blocks indicate either that the discretized level did not appear in the final decision ensemble or that it is a binary variable. In the decision network, nodes correspond to single variables and edges to interaction effects on the response between two nodes. Sizes represent feature and interaction importances, while colors describe feature and interaction influences. In addition, the edge type indicates the interaction direction, so that it is either a solid line if on average the pair of variables have the same sign (i.e., positively associated variables), or a dashed line, if not.

## Datasets

**Simulated datasets.** We generated $n$ independent observations of a random vector $(Y, K, V^1, \ldots, V^{12})$ as follows. Let $V^1, \ldots, V^{12}$ be independent $\mathcal{N}(0.5, 1)$ distributed random predictive variables, let $K$, a multiclass feature, be uniformly distributed over the categories {a, b, c, d}. The binary response $Y$ is set by the rules in Table 2, its sign is changed with a probability $r$ to add noise.

We used this data generating mechanism as a very simple model to evaluate our method as the underlying mechanism is fully understood here. For readers unfamiliar with abstract simulation settings, it may be helpful to think about the variables $V^1, \ldots, V^{12}$ as (re-scaled) microbial abundances, the categories $a$ to $d$ as phenotypes such as age groups and the response variable $Y$ as a disease indicator, with 1 and −1 encoding healthy and diseased, respectively. A single replicate of the data with parameters $n = 1000$ and $r = 0.05$ is given in S4(A)–S4(D) Fig.

When evaluating endoR on this simulated data (see Results) we used a RF model fitted with the randomForest R-package [43]. Details on the fitted model and the parameter settings of endoR are provided in the S2 Text.

**Artificial phenotypes.** To assess the performance of endoR under more realistic microbiome conditions, we additionally evaluated it on a real metagenomic dataset with a simulated response variable. We call this the *artificial phenotypes* data set, to stress that while the features are real metagenomic measurements, we artificially construct groups and a response variable, hence providing a known ground truth of the underlying model. The artificial phenotype designate the response variable.

We used the same collection of human gut metagenome datasets as in [115], with additional sample exclusion criteria and identical sequence processing (S2 Text). The dataset comprised

**Table 2. Predetermined decision rules to generate the response variable from the simulated datasets.**

| Decision rule | Response |
|---|---|
| Group = 'a' & V1>0 & V2>0 | 1 |
| Group = 'a' & V1≤0 & V2≤0 | 1 |
| Group = 'a' & V1>0 & V2≤0 | -1 |
| Group = 'a' & V1≤0 & V2>0 | -1 |
| Group = 'b' & V3>0 | 1 |
| Group = 'b' & V3≤0 | -1 |
| Group = 'c' & V4>0 & V5>0 | 1 |
| Group = 'c' & V4≤0 & V5≤0 | -1 |
| Group = 'd' & V6≤0 & V7>0 | -1 |
| Group = 'd' & V6>0 & V7≤0 | 1 |

2147 samples from 19 studies, with relative abundances of families, genera and species with a prevalence above 25% ($p$ = 520 taxa; S2 Text).

Based on these data, we artificially constructed a multi-class phenotypic variable $K$, uniformly distributed over the categories {$a$, $b$, $c$, $d$} for the replicate presented in Fig 2A–2E, or {$a$, $b$, $c$} otherwise. Within each group, combinations of randomly picked taxa with a prevalence higher than 50% were used to determine the sign of the response variable $Y$ (for the replicate in Fig 2, see Table 1 and Fig 2A–2E; otherwise, see the pooled list of rules from which decisions were drawn in S9 Table). Noise was added by changing the group label with a probability $r$: new group labels were drawn from {$a$, $b$, $c$, $d$, $e$} for the replicate in Fig 2A–2E, and from {$a$, $b$, $c$, $d$} otherwise. The procedure was repeated 51 times to generate the example presented in Fig 2A–2E, and the set of 50 APs was used otherwise.

We evaluated endoR on these artificial phenotypes based on fitted RFs and boosted trees, generated via the randomForest and xgboost R-packages, respectively [43, 46]. Models were generated by first applying a feature selection step and then fitting the classifier; cross-validation (CV) on 10 subsets was used to select the model (see S1(A) Fig for an overview of model training). All details on the fitted ML models and the parameter choices for endoR are provided in the S2 Text. Each model was processed with endoR using default parameters (i.e., $K$ = 2, $B$ = 10, and $\alpha$ = 5; Figs 3A–3C and 4, S4 and S8(F), S8(I), S8(J) Figs). For the replicate in Fig 2, numeric variables were discretized into 3 categories and $B$ = 100 bootstrap resamples.

The global null model for AP was generated with the same procedure as for the replicate in Fig 2A–2E, with the difference that target values were fully randomized within each group after being calculated and before adding noise by randomizing group labels. Consequently, the group structure was conserved, but the artificial phenotype was de-correlated from taxa relative abundances. The classifier was fitted as described above: we used 10x CV to select the best model consisting of a feature selection step and the fitting of a random forest classifier. The feature selection of the best model was very stringent, with only 4 features selected (expected accuracy = 56.27% with feature selection parameter gamma = 1 and classifier number of trees = 500). Therefore, we decided to also fit a slightly less accurate model that had a looser feature selection step (expected accuracy = 55.54%, feature selection gamma parameter = 0.35 and RF number of trees = 100), resulting in 44 selected features. Both models were interpretated via endoR with the same parameters as for the main replicate presented in Fig 2H–2J.

We similarly generated independent global null models for 10 of the artificial phenotypes used to evaluate parameters: feature selection was performed with small $\gamma$ values ($\gamma \in$ [0.2, 0.4]) and RF classifiers were fitted with 100 trees. On average, 130.8±29.83 features were selected, and the RF's expected accuracies were 0.59±0.05. endoR was applied to the 10 models with the same parameters as for the 50 independent APs.

**Cirrhosis metagenomes.** Metadata and gut microbial taxonomic profiles from metagenomes generated by Qin et al. [41] were downloaded from the MLRepo (https://github.com/knights-lab/MLRepo, accessed on 27/01/2021). The dataset consisted of 68 and 62 stools samples from cirrhotic and healthy individuals, respectively, for whom age, BMI and sex information were available (48% healthy individuals). The formatting of metagenomes and model fitting procedure are detailed in the S2 Text. In brief, rare taxa were filtered out before model fitting. An RF classifier with feature selection was fitted on the filtered taxa (see the S2 Text and S1(A) Fig for details on model training). The final model selected 85 of 926 taxa and certain metadata covariates (gender, age, BMI, and number of sequence reads). The model was processed using endoR with default parameters, except for the discretization into 3 categories, $B$ = 100 bootstrap resamples of size $3n/4$, and $\alpha$ = 5 (S2 Text).

The linear regression models with lasso penalty [53] were fitted using the `glmnet()` function from the glmnet R-package [116] on the same CV sets as for fitting the full RF model. The

lambda hyperparameter was tuned (parameter nlambda = 100 in the function). A model without interaction was trained on all features (BMI, age, gender, sequencing depth, and relative abundances at the family, genus, and species levels of the 926 taxa). We also trained a model on the same features and additionally included all pairwise interactions, bringing up the number of features to 429,201.

*Methanobacteriaceae* **metagenomes.** Metagenomes and associated sample metadata from a globally distributed set of studies were gathered from [40] by [115] (S2 and S3 Tables). Details on data processing are available in the S2 Text. Briefly, (i) metagenome were profiled with the HUMAnN2 pipeline to obtain metabolic pathways profiles based on the MetaCyc database [117, 118] and with Kraken2 and Bracken v2.2 based on a customized Genome Taxonomy Database (GTDB), Release 89.0 created with Struo v0.1.6 (available at http://ftp.tue. mpg.de/ebio/projects/struo/) [119–122] for the taxonomic profiles; (ii) rare taxa were filtered out and taxonomic ranks from family to species were included (*n* = 2190 taxa; 181 families, 562 genera and 1447 species; S17 Fig); (iii) relative abundances of MetaCyc metabolic pathways at the community level with complete coverage and a prevalence greater than 25% were included (*n* = 117 pathways). An RF with feature selection was trained to classify samples based on the presence/absence of *Methanobacteriaceae* using 10 CV sets (see the S2 Text and S1(A) Fig for details on model training, and S4 Table for the results of CV). The final model was processed with endoR with default parameters, except for *B* = 100 bootstraps and $\alpha$ = 5.

## Evaluation metrics and benchmark methods

**Simulated data.** A ground truth network was extrapolated from Table 2 (S4(E) Fig). The network constructed from the final decision ensemble by endoR (S4(H) Fig) was compared to the ground truth network by counting the numbers of true positive (TP), false positive (FP), and false negative (FN) nodes and edges.

**Artificial phenotypes.** Ground truth networks were extrapolated from the procedures used to create the artificial phenotypes (for an example, Table 1 corresponds to Fig 2G). Since the data set is made of real metagenomes, a deficit here was the lack of ground truth on associations among predictive variables, notably from the same taxonomic branch. Hence, to account for taxonomic relationships, we extended the lists of true nodes and edges to include nodes and edges from related taxa. We considered as 'related' taxa the direct coarser and finer taxonomic ranks, and species from the same genus. Consequently, a node identified by endoR was counted as TP if it was in the ground truth network, or related to a node in the ground truth network. If both a true node and a related taxon were identified by endoR, the TP was counted only once to prevent inflating results. The same counting was performed for edges.

**Metrics.** Based on the numbers of TPs, FPs, and FNs, standard performance metrics (accuracy, precision, recall) were calculated to evaluate networks generated by endoR. In addition, TPs and FPs were weighted by their feature or interaction importances (for nodes and edges, respectively) to calculate a weighted precision, and so estimate the magnitude of TP in the endoR results. Given a ranking over the decision importances, TP/FP curves could be constructed for nodes and edges. To do so with endoR, for a fixed $\alpha$, we first ranked the top *q* decisions of each bootstrap according to their probability of being selected in the final stable decision ensemble (i.e., the number of occurrences across bootstraps). Networks were computed for each probability of a decision to be selected, and the probabilities of edges and nodes to be in networks were subsequently calculated. Edges and nodes were then ranked by these probabilities and TP/FP curves were constructed for endoR (Fig 3A and 3D, S4, S8(C) and S8 (F) Figs). Curves were interpolated and averaged across repetitions.

**Comparison of endoR with state-of-the-art.** The comparison of endoR against state-of-the-art methods was based on the AP simulated data and consisted of the following steps (additional details are provided in the S2 Text). First, for all numeric variables in the dataset ($p = 520$ taxa), we performed a Wilcoxon-rank sum test to identify taxa enriched in samples labelled with one or the other response variable category ('-1' versus '1'), and we performed a $\chi^2$ test to assess whether group categories comprised more samples than expected from one or the other response category; $p$-values were adjusted using the Benjamini-Hochberg correction method; features were ranked by $1 - p$-value and effect size in case of ties. Second, we fitted two lasso models with lambda tuning on all variables in the dataset ($p = 520$ taxa and 4 one-hot encoded groups). For these models only, the relative abundances of taxa were transformed using the center-log ratio (see S2 Text). The first model did not include interactions (so only the variable main effects) and the second was fitted with all variables and pairwise interactions. Features were ranked based on their absolute weight in the final models; for the model with interactions, we used the sum of the main and interactions weights for each feature. Third, we divided samples according to their response variable category and used taxa relative abundances ($p = 520$ taxa) to build sub-networks for each category via the graphical lasso [123] and sparCC [47] methods, as implemented in the SpiecEasi R-package [124]; features were ranked by the square of covariance matrices parameters. For each method, edges shared between the two sub-networks were filtered out. From the RF model, we also computed Gini importances [19, 25], as implemented in the randomForest R-package [43], and SHAP values [35, 125], as implemented in the iBreakDown R-package [48]. We additionally trained an XGBoost model [46] on the same features selected by gRRF ($p = 18$ taxa and group dummy variables). The XGBoost model was trained with default parameters and nrounds = 10, objective = 'binary: logistic'. SHAP values and SHAP interaction values were extracted from it using the xgboost and SHAPforxgboost R-packages [46, 126], and it was finally processed with endoR. For Gini, SHAP and endoR, features and pairs of features were ranked by feature and interaction importances. Variables, or pairs of variables, were randomly drawn and sorted to build TP/FP curve; the process was repeated 1000 times and averaged.

## Bacterial genome analysis

Species-representative genomes from the GTDB-r89 database, which were used to obtain relative abundances of taxa in metagenomes [121], were downloaded for each species detected in the dataset before filtering. Genomes were annotated via DIAMOND blastp [127] against the following databases: (i) Fungene [82] to identify the *dsrA* and *dsrB* genes, (ii) hydDB [84] to identify genes coding for hydrogenases, and (iii) acetobase [83], to identify the *fhs* gene. The *dsrA* and *dsrB* genes encode disulforedoxins involved in sulfate-reduction and the *fhs* gene encodes the formyltetrahydrofolate synthetase involved in acetogenesis. Hydrogenases were grouped by predicted function: $H_2$-production, $H_2$-uptake, bidirectional, and sensory (S5 Table). For each species, the number of gene copies with a percent sequence identity above 0.50 and a length coverage above 80% was counted. For genus and family taxonomic ranks, the number of copies were averaged across species and weighted by the average relative abundance of each species in the dataset used for analysis. Patterns of gene abundances observed with absolute copy numbers were robust to differences in genome size (S15(A) Fig).

The gene set enrichment analysis was performed using the fgsea R-package [88]. Taxonomic features were used as 'genes' and ranked by Gini or endoR feature importance, and 'gene sets' were defined by gene group (Acetogen, SRB, and hydrogenases predicted functions).

## Statistics

All statistical analyses were performed in R using the stats package [128]. For the *Methanobacteriaceae* analysis, we measured the over-representation of taxonomic orders in the set of features used by endoR using a Monte-Carlo procedure (S8 Table). For this, we approximated the number of family, genus, and species features expected by random, given the number of features used by endoR, by randomly drawing eighteen features from the set of taxonomic features used to fit the model. We repeated the random draws 1000 times. For each draw, the number of features belonging to each order was counted. The null distribution of each order was obtained by pooling all counts across draws, and the right-tailed *p*-value of the observed count was calculated from this null distribution.

## Supporting information

**S1 Text. Supplementary Methods describing the endoR method.**
(PDF)

**S2 Text. Supplementary Methods describing the data, the evaluation of endoR, and the analysis of metagenomes.**
(PDF)

**S3 Text. Supplementary Results.**
(PDF)

**S4 Text. Abbreviations.**
(PDF)

**S1 Fig. Model selection and fitting for predicting the presence/absence of *Methanobacteriaceae*.** A/ Ten sets of observations, each containing a subset for training and one for testing, were created. Training observations were used to fit models, i.e., the combination of a feature selection and classifier algorithms for given hyperparameter values, and testing ones were predicted with the fitted models. Model's performances were averaged across testing sets. Feature selection algorithms consisted of (i) no feature selection, (ii) a taxa-aware version of the gRRF algorithm [21] (S2 Text), (iii) the Boruta algorithm [24], and (iv) no feature selection. Classifiers were fitted with random forests or gradient boosted model algorithms. Metadata correspond to the number of reads and original dataset names. B/ The model (feature selection algorithm and classifier) that resulted in A/ in the highest average Cohen's $\kappa$ using the fewest features was used to fit the final classifier on all data.
(PDF)

**S2 Fig. endoR recovers ground truth network from perfect predictive models.** endoR was applied to the set of rules directly obtained from the true mechanism generating the response variables for one replicate of each of the AP (A-B) and FSD (C-D) simulations. No regularization step was performed, i.e., no pruning nor bootstrapping. Respective ground truth networks are visualised in Fig 2F and S4F Fig. The additional edges on B are due to the discretization step. No additional edge appears on D due to the proximity between the median of numeric features (used to discretize data) and the thresholds used to make the response variable.
(PDF)

**S3 Fig. endoR generally does not find stable decision ensemble from global null models.** Global null models were generated from 10 APs by randomizing the target values within each group (see Methods). A predictive model was then fitted, including a FS step followed by the fitting of a RF classifier. Models were interpreted with endoR: no stable decision ensemble was

reached in 6/10 cases, a unique stable decision was found in 3/10 cases, and a stable decision ensemble was found for the replicate which had the highest RF accuracy.
(PDF)

**S4 Fig. endoR captures interactions predictive of a response variable from a random forest fitted on simulated data.** A-D/ Fully simulated data (FSD) structure: four groups of samples (labelled from a to d) were generated so that for each group, the binary response variable takes the value '1' (blue) or '-1' (yellow) according to a combination of variables described in Table 1 (e.g., V1 and V2 for Group a). The values of the response variable were then randomized with a probability $r = 0.05$. E/ Ground truth network of associations between the response variable and single variables (nodes) and pairs of variables (edges) described in A/ (see Methods). Pairs of variables predicting '1' are linked by a blue edge ('positive') and those predicting '-1' by a yellow edge ('negative'). Variables for which high values are predictive of '1' have a blue node color ('positive') and a yellow node color if high values are predictive of '-1' ('negative'). If high values are predictive of '1' or '-1' depending of other variable values (e.g., Group b predicts '1' if V3 takes high values, but '-1' if V3 has low values), the color is grey ('depends'). F/ Feature importance as measured by the mean decrease in Gini impurity in the fitted random forest (RF) model trained on the dataset shown in A/. G/ Feature importance as measured by endoR and feature influences for each discretized level of numeric variables as computed by endoR. The point color indicates whether features were used to construct the response ('True') or not ('Irrelevant'). H/ Decision network produced by endoR. Edges and nodes correspond to single variables and their interaction effect on the response variable, respectively. Edge widths and node sizes are proportional to the interaction and feature importances calculated by endoR, respectively; their colors are representative of their influence (see Methods in main text and S2 Text for details on network construction). The edge transparency is inversely proportional to the importance for H only. I/ Same than H but edges with lowest interaction importance were removed to obtain paths between nodes of length $\leq 3$. E, G-I/ Levels of discretized variables, i.e., numeric variables transformed into categorical variables based on their quantiles, are shown on the x-axis of the influence plot (G/) and indicated by '__High' or '__Low' in networks (E/ and H/).
(PDF)

**S5 Fig. The accuracy of endoR increases with the accuracy of input models.** 100 FSDs (B-E and G-H/) and 50 APs (A, F, and I-J/) were generated, RFs were fitted and processed with endoR. If not varied, parameters were as follow: *ntree* = 500, discretization was performed with the method based on the data distribution with $K = 2$ categories, and $\alpha = 5$. We computed the following three metrics: Cohen's $\kappa$ of the RF, weighted precision and recall values of the selected edges in the stable decision ensemble, and TP/FP-curves based on the probabilities of being selected in the stable decision ensemble (see Methods). A-B, D-E, and G-J/ TP/FP-curves are averaged across all datasets for a fixed parameter setting (line) and standard deviation (shaded area) are displayed. The average number of TPs and FPs expected for a randomization null model and standard deviations, are shown in grey. Large points indicate the average number of TPs and FPs in the stable ensembles generated by endoR. C and F/ Each point corresponds to the precision/recall of endoR applied to a single dataset and parameter setting. The larger traced points are the averages across all datasets for a fixed parameter setting. A-B and D-E/ As expected decreasing the noise or increasing the number of trees in the forest improves the performance of endoR both in terms of precision and recall. Importantly, there is a strong dependence of endoR performance on the performance of the fitted RF and endoR. Moreover, endoR has a good precision even for small RF. C/ Increasing $\alpha$ increases both the TPs and FPs. Small values of $\alpha$ effectively control the FPs without strongly impacting the recovered TPs. F/

Larger values of *B* are slightly better but endoR performs well even for small values of *B*. G-J/ Discretization was performed by creating K = 2 or 3 categories from numeric variables based on their distribution ('data') or on the splits on these variables in the fitted RF ('RF thr'). Discretization slightly influences endoR performance, without any clear pattern between the FSD and AP simulations.
(PDF)

**S6 Fig. endoR performance stabilizes as the number of bootstraps increases.** A total of 6 replicates of artificial phenotypes were each processed 10 times with *B* = 10 or 100 bootstraps resamples (purple and orange, respectively). The curves show the average number ('#') of identified true positive (TP) and false positive (FP) edges according to edge probabilities of being selected in the stable decision ensemble. Curves were interpolated for each technical replicate, and the average (line) and standard deviation (shaded area) across number of bootstraps are displayed. The traced points denote average number of TP and FP in the stable ensembles returned by endoR for $\pi = 0.7$ and $\alpha = 5$.
(PDF)

**S7 Fig. endoR computation time scales substantially better than SHAP when applied to random forest classifiers.** A-F/ The total CPU time and maximal virtual memory used for three replicate processing runs of the same RF model. The artificial phenotype presented in Fig 2 was used with 18 variables and 1000 samples (see Methods), and endoR was run on *B* = 1 bootstrap of size *n*/2. G/ Five technical replicates of endoR and shap runs on the RF trained to predict the artificial phenotype presented in Fig 2 (18 variables and 2147 observations). Calculations were ran in parallel across 4 or 10 workers; for endoR, bootstraps were further ran individually in parallel (see S2 Text).
(PDF)

**S8 Fig. Discretization of variables and modification of rules.** Simple example of the discretization of a uniformly distributed variable *x* into three levels. An original rule "*x* < *t*" (orange) is modified according to the number of observations in each level included in the sample support of the rule (new rule-s in green). B/ A minority of samples in the "Medium" level were included in the original sample support defined by "*x* < *t*", therefore the "Medium" level is not selected to make a new rule as in C/.
(PDF)

**S9 Fig. endoR performs as well as state-of-the-art methods at identifying variables and pairs of variables predictive of a target from fully simulated data.** Average (line) and standard deviation (area) of identified true positive (TP) for a given number of false positive (FP). The average numbers of TP and FP in the endoR final decision ensemble are indicated with points. A, C: correspond to single variables and B, D: to pairs of variables across 100 replicates of fully simulated data. A, B/ the truncated lines of absolute numbers of TP and FP are displayed, dashed grey lines denote the ground truth number of TP. C, D/ the full curves of TP and FP rates are displayed. Lines are dashed when necessary due to overlaps. 'Random' signifies results expected with a randomization null model. A, C/ All methods almost identified only TP first and then FP. B, D/ endoR better discriminated TP from FP edges than SHAP. Only endoR does not return all features and interactions, hence limiting the number of FPs in the final decision ensembles, although resulting in lower recall too.
(PDF)

**S10 Fig. SHAP values from the RF classifier.** SHAP values were calculated from the random forest classifier trained to predict an artificial phenotype simulated from real metagenomes

($n$ = 2147, $p$ = 520 taxa; see Fig 2). A/ The feature importance is given by the average of the absolute SHAP values across samples and is plotted for each sample as well. B/ The SHAP interaction values could not be calculated due to the lack of R-implementation for calculating SHAP interactions from random forests. Consequently, we plotted SHAP values of the four taxa with the highest feature importances (y-axis) according to taxa relative abundances (log10 transformed, x-axis) and colored points by each of the four group category (pink: sample from the group category indicated in the plot title, blue: sample from the other categories). We note that this method of analysis does not scale well as the number of features increases.
(PDF)

**S11 Fig. SHAP values from the XGBoost classifier.** SHAP values were calculated from the XGBoost classifier trained to predict an artificial phenotype simulated from real metagenomes ($n$ = 2147, $p$ = 520 taxa; see Fig 2). A/ The feature and interaction importances are given by the average of the absolute SHAP values across samples. B/ Given the high number of features and interactions, we only plotted the top five feature importances of single variables and top nine feature importances for interactions (marked with a start on A/). For single variables, the point color corresponds to the x-axis value.
(PDF)

**S12 Fig. Relative abundances of taxa identified using a RF and endoR versus statistical tests in the original study [41].** When the log10 of relative abundances is displayed, a pseudo-count equal to the minimal relative abundance detected in the dataset ($3 \cdot 10^{-7}$) was used to show samples for which taxa were not detected (relative abundance = 0). Boxplots and points are colored by healthy status, with healthy individuals in orange and cirrhotic ones in blue. A/ Taxonomic levels are indicated with the prefixes: 'f_' = family, 'g_' = genus, 's_' = species. Taxa are organized by family taxonomic level (separated by grey lines). The background indicates whether taxa were identified in this article and the original study (red), only in this article via a RF model and endoR (green), or only in the original study (yellow). Species for which the relative abundances were not available in the published dataset (downloaded from the ML task repository) are indicated with a star. Taxa in B-I/ are indicated by an arrow. B-E/ The four taxa taxa with highest feature importance (FI) identified by endoR to classify healthy versus cirrhotic microbiomes (see Fig 5A). F-I/ Taxa exclusively identified in the original study [41] or with a random forest and endoR.
(PDF)

**S13 Fig. The Gini and endoR importances are consistent between the RF model and stable decision ensemble.** A/ Features with the best Gini importance. B/ Comparison of the Gini importance and the number of times features were selected across the 10 cross-validation (CV) sets. C/ Comparison of the Gini and endoR importance. D/ Gini importance of all features selected by the taxa-aware gRRF algorithm. In all plots taxonomic levels are indicated in the labels with 'f_': family, 'g_': genus, and 's_': species, and orders are indicated via point and label colors.
(PDF)

**S14 Fig. The relative abundance of *Clostridia* is higher in samples where *Methanobacteriaceae* are detected.**
(PDF)

**S15 Fig. Copy number of genes involved in H$_2$ consumption and production across taxa used to predict the presence/absence of *Methanobacteriaceae*.** We looked into the genomes of representative species of taxa used to predict the presence/absence of *Methanobacteriaceae*

in human guts microbiome from 2203 individuals for involved in $H_2$ metabolism. The number of copies of genes involved in the following pathways or function were counted: sulfate reduction (SRB): *dsrA* and *dsrB* genes [82]; acetogenesis (Acetogen): *fhs* gene [83]; $H_2$ production, uptake and sensing as determined by the HydDB database [84]. At the genus and family taxonomic levels, we used the average number of copies across species from the given level and weighted the number of copies of each species by the average relative abundance of species in the dataset. Accordingly, if the most abundant species of a specific genus had high number of gene copies, the number of copies for that genus would also be high. When genes were grouped by general function, we summed the number of copies (e.g., the SRB gene copy number corresponds to the sum of gene copies of *dsrA* and *dsrB*). A/ The ratios of gene copy number by genome size for each of the endoR selected features are consistent with the absolute number of copies displayed in Fig 6B. For each representative species, the number of gene copies was divided by the genome size. General functions and genes are displayed and separated by black lines (blocks of genes with the same general function), general functions are separated from specific genes by a grey line. B/ Number of genes copies from each group for taxa selected by feature selection. C/ Occurrence of genes from each group across all taxonomic features used to train models to predict the occurrence of *Methanobacteriaceae* in human guts. (PDF)

**S16 Fig. Visualization of how decisions are modified to calculate the importance of variables.** Each plot illustrates the support of a decision $D$ in the feature space spanned by variables $\{x^j, x^k\}$, i.e., the values that the decision can take on variables $x^j$ and $x^k$. A/ Original decision $D$. B/ Modified decision $D_j^{rm}$ resulting from removing variable $x^j$ from decision $D$. C/ Modified decision $D_{j,k}^{rm}$ resulting from removing variables $x^j$ and $x^k$ from decision $D$. A-C/ The support $S_D$ of the originl decision is indicated by the stripped areas, such as samples in the support of $D$ all take positive values on $x^j$ and $x^k$. The support of each decision, i.e., $S_D$, $S_{D_j^{rm}}$ and $S_{D_{j,k}^{rm}}$ for A, B, and C, respectively, is visualized by the colored region. B/ When we remove variable $x^j$ from the rule $r_D$ of $D$, the support $S_{D_j^{rm}}$ is extended to samples taking negative values on $x^j$ (colored area). C/ Similarly, when we remove a pair of variables $\{x^j, x^k\}$ from $r_D$, samples in $S_{D_{j,k}^{rm}}$ can take positive and negative values on $j$ and $k$. For $S_{D_j^{rm}}$ and $S_{D_{j,k}^{rm}}$, we calculate $\hat{y}_{D_j^{rm}}$ and $\hat{y}_{D_{j,k}^{rm}}$, respectively, using all samples in $S_{D_j^{rm}}$ and $S_{D_{j,k}^{rm}}$. The decision-wise importance $\delta_D^j$ of $j$ in $D$ is calculated by comparing the error of $\hat{y}_{D_j^{rm}}$ on $S_D$ (B/) versus the error of $\hat{y}_D$ on $S_D$ (A/). Similarly, to calculate the decision-wise importance of a pair of variables $\{j, k\}$ in a decision $D$, we compare the error from the decision not constraining values on $j$ or $k$, with $\hat{y}_{D_{j,k}^{rm}}$ on $S_D$ (C/) to the error of the decision with $\hat{y}_D$ on $S_D$ (A/). (PDF)

**S17 Fig. Mean relative abundances and prevalence of family, genus, and species taxonomic levels in the metagenomic data.** (PDF)

**S18 Fig. Gut microbiota of a large cohort of healthy individuals (n = 2203 individuals) approximately segregate along the enterotype landscape.** A-C, H-K/ Principal coordinate analysis ordination of the Jensen-Shannon distance matrix computed from genera relative abundances. A-B/ Samples colored by relative abundance (RA) of *Bacteroides* and *Prevotella*, respectively. To calculate the log, a pseudocount equal to the minimal non-null RA was given to samples for which the genus was not detected, i.e., RA = 0. C/ Samples colored by enterotype

cluster [2, 64]; ETF: *Firmicutes*, ETB: *Bacteroides*, ETP: *Prevotella*; colors correspond to those on E. J-K/ Samples are colored by their country of origin grouped by region when possible (e.g., Canada and USA grouped into North America; S7 Table), and points are emphasized (larger and less transparent) if they were sampled from non-westernized (J) or westernized (K) populations. D-G/ Average silhouette score (bar) within each k-mean cluster computed from the Jensen-Shannon distance matrix and across clusters (thick line). Dashed line: threshold above which clustering strength is moderate.
(PDF)

**S1 Table. Average random forest accuracy and network metrics from the simulations on toy datasets.**
(CSV)

**S2 Table. Datasets and country of origins of samples used for associating gut bacterial features to the presence of *Methanobacteriaceae* in human guts.**
(CSV)

**S3 Table. Available sample metadata for associating gut bacterial features to the presence of *Methanobacteriaceae* in human guts.** Summary: For numeric variables: minimal—maximal values (median and mean ± standard deviation). For categorical variables: each level (number of samples in the level). Region: Regions group samples from countries of a same geographic area. Countries from unique region are designated with their country name to prevent confusion.
(CSV)

**S4 Table. Predictive performance of models trained to predict the presence of *Methanobacteriaceae* from metagenomes.** Samples: Models trained on CV sets from all samples ($n$ = 2203 samples in total, train = 1542 and test = 661 samples), or only the set of samples with complete metadata information for age, gender, and BMI ($n$ = 748 samples in total, train = 524 and test = 224 samples). Feature selection: Feature selection algorithm and its tuned parameters Model: Random forests (RF) were fitted using the ranger R-package [44]; ntrees = 250 and 500 were tested for parameter tuning. Gradient boosted models (XGBoost) were fitted using theXGBoost R-package [46]; nrounds in {10, 50, 100, 250, 500, 750, 1000, 1500} and maxdepth in {1, . . .,10} were tested for hyperparameter tuning (only the results of the model with highest average Cohen's kappa are given). Sample weights were provided to increase the probability of samples from the under-represented class to be sampled at each bootstrap. Accuracy, Cohen's kappa, and number of selected features: Average +/- standard deviation across 10x cross-validation 70–30% train-test sets. The selected model is indicated in bold.
(CSV)

**S5 Table. Genes sought in species representative genomes of microbial features used to predict the occurrence of *Methanobacteriaceae* in human gut microbiome.**
(CSV)

**S6 Table. Stable decision ensemble extracted by endoR from the model predicting the presence/absence of *Methanobacteriaceae* in human gut microbiomes.**
(CSV)

**S7 Table. Importance and influence of single variables extracted from the stable decision ensemble in S6 Table. The taxonomy is also given [121].**
(CSV)

**S8 Table. Count of features, grouped by taxonomic order, used by endoR to predict the presence of *Methanobacteriaceae* and right-tailed p-values from Monte-Carlo procedure of being used at this frequency.**
(CSV)

**S9 Table. List of all predetermined rules for making the response variables from metagenomic data for artificial phenotype simulations.**
(CSV)

**S10 Table. Average random forest Cohen's $\kappa$ and network metrics from the artificial phenotypes.**
(CSV)

**S11 Table. Computation time and memory of endoR and the iBreakDown R-package to compute SHAP.**
(CSV)

**S12 Table. Stable decision ensemble extracted by endoR from the RF model trained to classify samples from the Qin et al. [41] dataset according to metadata and relative abundances of taxa selected using our modified version of the gRRF feature selection algorithm.**
(CSV)

**S13 Table. Importance and influence of single and pairs of variables extracted from the stable decision ensemble in S12 Table.**
(CSV)

## Acknowledgments

We thank Sofia Esquivel-Elizondo for the discussions about hydrogenases and methanogens, and Jacobo de la Cuesta, Daphne Welter, and Brandon Seah for their feedback on the manuscript. We also thank the reviewers for their insights. We are grateful to all who make their data open and/or who contribute to open science. Open sharing of data enabled this project.

## Author Contributions

**Conceptualization:** Albane Ruaud, Niklas Pfister, Nicholas D. Youngblut.

**Data curation:** Albane Ruaud, Nicholas D. Youngblut.

**Formal analysis:** Albane Ruaud, Nicholas D. Youngblut.

**Funding acquisition:** Ruth E. Ley.

**Investigation:** Albane Ruaud.

**Methodology:** Albane Ruaud, Nicholas D. Youngblut.

**Project administration:** Niklas Pfister, Ruth E. Ley.

**Resources:** Nicholas D. Youngblut.

**Software:** Albane Ruaud.

**Supervision:** Niklas Pfister, Ruth E. Ley, Nicholas D. Youngblut.

**Validation:** Albane Ruaud.

**Visualization:** Albane Ruaud, Nicholas D. Youngblut.

**Writing – original draft:** Albane Ruaud, Niklas Pfister, Nicholas D. Youngblut.

**Writing – review & editing:** Albane Ruaud, Niklas Pfister, Ruth E. Ley, Nicholas D. Youngblut.

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
