## [Decision Letter · Decision Letter 0]

9 Jun 2022

Dear Dr. Youngblut,

Thank you very much for submitting your manuscript "Interpreting tree ensemble machine learning models with endoR" for consideration at PLOS Computational Biology.

As with all papers reviewed by the journal, your manuscript was reviewed by members of the editorial board and by several independent reviewers. In light of the reviews (below this email), we would like to invite the resubmission of a significantly-revised version that takes into account the reviewers' comments.

The reviewers generally consider that the work is a good contribution to the field and the manuscript is well-written. However, they also raise some points that should be addressed. In particular, the concern of reviewer #1 as to how the feature selection is performed is most worrisome as it is an issue of correctness.

We cannot make any decision about publication until we have seen the revised manuscript and your response to the reviewers' comments. Your revised manuscript is also likely to be sent to reviewers for further evaluation.

Sincerely,

Luis Pedro Coelho

Associate Editor

PLOS Computational Biology

Dina Schneidman

Software Editor

PLOS Computational Biology

The reviewers generally consider that the work is a good contribution to the field and the manuscript is well-written. However, they also raise some points that should be addressed. In particular, the concern of reviewer #1 as to how the feature selection is performed is most worrisome as it is an issue of correctness.

Reviewer's Responses to Questions

**Comments to the Authors:**

Reviewer #1: Ruaud and colleagues present endoR, an R package aimed to enable better interpretation of tree ensemble methods, particularly for microbiome data. They showcase the utility of their package using fully simulated data, artificial phenotypes (real data with a simulated response variable), and two real-world applications (prediction of liver cirrhosis from fecal metagenomes and prediction of the presence of Methanobacteriaceae in the gut). The software addressed a well-known problem in the field (i.e. limited interpretability of complex tree-ensemble machine learning methods) and their article is well written (albeit a bit verbose in places), but several questions remains, especially regarding feature selection, comparisons to other tools/measures and comparisons to other machine learning methods.

Feature selection: It sounds like the feature selection step was performed before model training (I hope I did not miss a detailed description of the feature selection, but in any case I would advise the authors to describe their feature selection procedure more prominently and in more detail). This can lead to overfitting and over-optimistic models and should be avoided (see for example https://www.ncbi.nlm.nih.gov/pubmed/19880370).

Comparison to other methods: The comparison to other methods (Gini coefficient, SHAP values, etc.) is the part of the manuscript that could be improved the most. Figure 2G indicates that endoR might perform better than the Gini coefficient, but it is only shown for the top features in the main text. Similarly, Figure 4 focuses on the top 5 false positive features only (disregarding the reconstructed network for now). I think it would be possible to construct AUROC measures for how well Gini coefficients, SHAP values, and endoR can recover the true underlying features (in repeated(?, see below) AP simulations), covering all features by evaluating the true positive and false positive rates, even though closely related features would have to be disregarded or evaluated separately.

Comparison to other machine learning methods: There are also other machine learning algorithms except the random forest routinely used in the microbiome field. For example, both LASSO or Elastic Net can have similar (or even slightly better) performance to the random forest (https://pubmed.ncbi.nlm.nih.gov/33785070/, given properly normalised input data), and both of these do not suffer of limited interpretability. The authors should compare their endoR importance measures also against features weights derived from those more white-box models.

Limited interactions in cirrhosis predictions: For the cirrhosis dataset, almost no meaningful interaction between features has been detected by endoR. I think the authors should discuss this observation. Does this mean that interaction networks between features do not play a prominent role in disease predictions? Relating to the point above, would a less-complex machine learning algorithm (such as LASSO) have been sufficient? (This is probably outside the scope of this paper, but how do different diseases compare in this aspect?)

Minor questions:

The prevalence threshold of 25% seems rather high. Can the authors motivate the level of the cutoff? For example, one of the best-known predictors for colorectal cancer, Fusobacterium nucleatum, is not present at a prevalence higher than 10% in a typical case-control study, and therefore, a prevalence cutoff of 5% might be more appropriate.

The software described in this article is available on Github. Are there plans to submit the R package to the Bioconductor repository? Also, the existing vignettes are rather short and do not include many details. Can those be expanded for better usability of the software?

Was the creation of the artificial phenotype repeated several times? It seems that the fully simulated data were generated with 100 independent repetitions, but I could not find this information for the artificial phenotypes.

The references to Figure 4 on page 7 are probably referring to Figure 5 and should be adjusted

The asterisk glyphs in Figure 6E are explained in the figure legend but could also be explained in a visual figure legend (such as in Figure 6C).

Reviewer #2: In this manuscript, the authors developed an algorithm for interpreting the results from those tree-based machine learning algorithms. Tree-based ML methods have been widely used for microbiome-based prediction due to the complex characteristics of microbiome data and potential high-order interactions among the taxa. However, these ML methods often did not provide sufficient interpretation. Thus, the work is considered to be very significant. The manuscript also reads well and the experiments are carefully designed to demonstrate the method. I have some comments the authors may consider in revision.

1. The interaction importance defined as the square root of the product of individual feature importance did not seem to conform to the traditional statistical interaction, which describes the phenomenon that the effect of one explanatory variable on the outcome depends on another explanatory variable or the joint effect of two explanatory variables is different from the sum of individual effects. Given the current definition, I do not see how this is related to the traditional statistical interaction. Please elaborate.

2. How does the algorithm perform under small sample sizes? In the real world, the majority of microbiome studies have small sample sizes. Performance under small sample sizes should be studied and caveat should be given if the algorithm does not work well for small sample sizes.

3. Global null conditions should also be studied. Given that there is no relationship between the microbiome and the phenotype, what is the frequency the algorithm will return any findings, and how many false positives if making any findings?

4. In the analysis of Methanobacteriaceae phenotype, how to address the compositional effects? Since the sequencing data are compositional in nature, the increase in the abundance of one taxon will naturally lead to decreases in the relative abundances of other taxa so there is a natural negative influence on the relative abundance of Methanobacteriaceae.

5. The bootstrapping the authors describes seems to be the sub-sampling technique, not sampling with replacement.

6. It does not make much sense to compare to those network inference algorithms such as sparCC and gLasso since they are detecting different networks.

Reviewer #3: The paper by Albane Ruaud et al. presents endoR, a machine learning strategy aiming at interpreting a fitted tree ensemble model. The method is validated experimentally on both simulated and real metagenomic data. Results show that endoR is able to infer true associations with more or comparable accuracy than other state-of-the-art approaches while easing model interpretation. The software is available as an open-source R-package on GitHub (https://github.com/leylabmpi/endoR).

This is a nice contribution to the microbiome community. The manuscript is well written and structured.

I have few minor comments:

- Is the method suitable for (shotgun) metagenomic data only or could be adopted also for 16S data? Please specifiy better this aspect in the text.

- I understand that data used for the analysis are too big to be hosted in github. Could you please put them somewhere else?

**Have the authors made all data and (if applicable) computational code underlying the findings in their manuscript fully available?**

Reviewer #1: Yes

Reviewer #2: None

Reviewer #3: **No: **Please make data used for this analysis public available

PLOS authors have the option to publish the peer review history of their article (what does this mean?). If published, this will include your full peer review and any attached files.

Reviewer #1: No

Reviewer #2: No

Reviewer #3: No
---

## [Decision Letter · Decision Letter 1]

22 Oct 2022

Dear Dr. Youngblut,

Thank you very much for submitting your manuscript "Interpreting tree ensemble machine learning models with endoR" for consideration at PLOS Computational Biology. As with all papers reviewed by the journal, your manuscript was reviewed by members of the editorial board and by several independent reviewers. The reviewers appreciated the attention to an important topic. Based on the reviews, we are likely to accept this manuscript for publication, providing that you modify the manuscript according to the review recommendations.

We believe that major issues have been addressed and that the outstanding doubts of Reviewer #1 are quickly addressable.

Sincerely,

Luis Pedro Coelho

Academic Editor

PLOS Computational Biology

Dina Schneidman

Software Editor

PLOS Computational Biology

We believe that major issues have been addressed and that the outstanding doubts of Reviewer #1 are quickly addressable.

Reviewer's Responses to Questions

**Comments to the Authors:**

Reviewer #1: I want to thank the authors very much for addressing my previous comments. I can imagine that it was a lot of work to include the LASSO classifier in their comparison as well and I appreciate their effort. Also, I thank the authors to submitting their package to CRAN. I hope it can be useful to the wider community. Lastly, restructuring the paper improved its readability very much.

I would have some question regarding the comparison to other methods, again.

Supplementary Figure 7 and Supplementary Figure 8 EF seem to identical. Is this intended? Also, I cannot really find correspondence between Figure 4 and Supplementary Figure 8 A-D, although they should show the same data (evaluation on the AP data). I would hope that the authors can clarify my confusion here. For example, the orange dot in SFig 8A indicated that at this FPR (it should be FPR in the axis label as well), only LASSO with interactions and the Wilcoxon perform worse than endoR, but in Fig 4A, Lasso with interaction performs better than the Wilcoxon and normal Lasso performs worse (but has a similar TPR in SFig 8A). From the figure legends I would think that the same data are shown, but as rates instead of numbers. Also, I would suggest to show both rates and the numbers in the main figure. As a reader, I personally would be more interested into the TPR at a given FPR than in absolute numbers for five FP detections.

Reviewer #2: The authors have addressed my comments sufficiently.

Reviewer #3: I feel the authors have answered positively to my previous comments and modified the manuscript according to them.

**Have the authors made all data and (if applicable) computational code underlying the findings in their manuscript fully available?**

Reviewer #1: Yes

Reviewer #2: Yes

Reviewer #3: Yes

PLOS authors have the option to publish the peer review history of their article (what does this mean?). If published, this will include your full peer review and any attached files.

Reviewer #1: No

Reviewer #2: **Yes: **Jun Chen

Reviewer #3: No

Figure Files:

Data Requirements:

Reproducibility:

References:

---

## [Editor Report · Decision Letter 2]

7 Nov 2022

Dear Dr. Youngblut,

We are pleased to inform you that your manuscript 'Interpreting tree ensemble machine learning models with endoR' has been provisionally accepted for publication in PLOS Computational Biology.

Best regards,

Luis Pedro Coelho

Academic Editor

PLOS Computational Biology

Dina Schneidman

Software Editor

PLOS Computational Biology

---

## [Editor Report · Acceptance letter]

8 Dec 2022

PCOMPBIOL-D-22-00408R2 

Interpreting tree ensemble machine learning models with endoR

Dear Dr Youngblut,

I am pleased to inform you that your manuscript has been formally accepted for publication in PLOS Computational Biology. Your manuscript is now with our production department and you will be notified of the publication date in due course.

With kind regards,

Anita Estes
